# Image Tokenizer Needs Post-Training

## Abstract

Recent image generative models typically capture the image distribution in a pre-constructed latent space, relying on a frozen image tokenizer. However, there exists a significant discrepancy between the reconstruction and generation distribution, where current tokenizers only prioritize the reconstruction task that happens before generative training without considering the generation errors during sampling. In this paper, we comprehensively analyze the reason for this discrepancy in a discrete latent space, and, from which, we propose a novel tokenizer training scheme including both main-training and post-training, focusing on improving latent space construction and decoding respectively. During the main training, a latent perturbation strategy is proposed to simulate sampling noises, i.e., the unexpected tokens generated in generative inference. Specifically, we propose a plug-and-play tokenizer training scheme, which significantly enhances the robustness of the tokenizer, thus boosting the generation quality and convergence speed, and a novel tokenizer evaluation metric, i.e., pFID, which successfully correlates the tokenizer performance to generation quality. During post-training, we further optimize the tokenizer decoder regarding a well-trained generative model to mitigate the distribution difference between generated and reconstructed tokens. With a ∼400M generator, a discrete tokenizer trained with our proposed main training achieves a notable 1.60 gFID and further obtains 1.36 gFID with the additional post-training. Further experiments are conducted to broadly validate the effectiveness of our post-training strategy on off-the-shelf discrete and continuous tokenizers, coupled with autoregressive and diffusion-based generators.

## 1 Introduction

In recent years, image generative modeling has been dominated by two major paradigms: diffusion (Dhariwal & Nichol, 2021; Song et al., 2022), which operates on the continuous latent space (Peebles & Xie, 2023; Rombach et al., 2022; Vahdat et al., 2021; Nichol & Dhariwal, 2021) through denoising, and autoregressive (AR) (Van Den Oord et al., 2016), which relies on the discrete latent space for next token prediction. Image tokenizers have emerged as an essential component to convert the raw pixels into continuous and discrete representations for diffusion and AR model, respectively.

Tokenizers aim to provide highly compressed yet structurally meaningful representations for downstream generative models (Song et al., 2022; Van Den Oord et al., 2016), substantially improving both the efficiency and scalability of training large generative models. A series of works has further advanced tokenizer design, introducing various improvement directions such as reconstruction quality (Chen et al., 2025b; Kim et al., 2025), compressed ratio (Chen et al., 2024b; Yu et al., 2024c), quantization method (Yu et al., 2023b; Lee et al., 2022a), and latent regularization (Kingma & Welling, 2013; Li et al., 2024c).

However, although these improvements have been validated under reconstruction metrics, their effectiveness under generative settings remains more of a black box. In practice, we often observe that tokenizers with strong reconstruction ability do not necessarily yield high-quality generations, whereas others with weaker reconstruction performance surprisingly lead to better generation results. This performance discrepancy between reconstruction and generation exposes a fundamental gap in how tokenizers are utilized across reconstruction and generation tasks. As illustrated in Figure 1 (a), generative modeling, including both AR and diffusion, inevitably introduces a distribution difference for tokenizers between generative training and sampling. Specifically, during training,

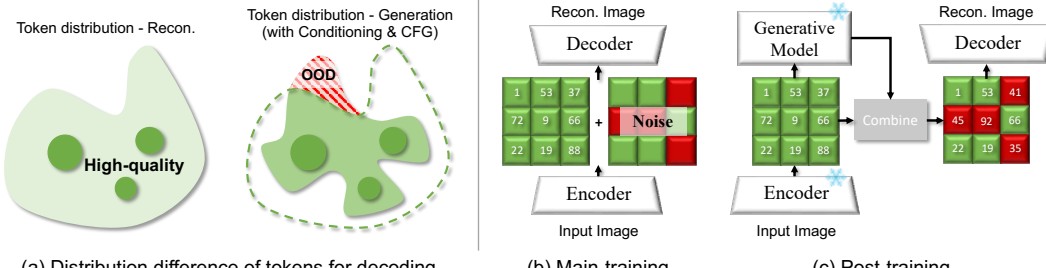

Figure 1: (a) Discrepancy between reconstruction and generation task imposes a latent token distribution difference between them. Specifically, reconstruction always rely on true tokens whereas generation task always sample out-of-distribution (OOD) tokens. To resolve this problem, we propose RobusTok (b) to enhance the robustness of tokenizer during main-training by latent perturbation, and (c) align the generated latent space with its target image in post-training stage.

ground truth representations are provided (Sutton, 1988; Song et al., 2022), and during sampling, future predictions can only rely on previous predicted ones.

This issue is further amplified by the typically misaligned objectives of tokenizer training and generator sampling. Tokenizer training prioritizes reconstruction fidelity where the visual decoder takes *clean* image tokens for accurate image reconstruction. Instead, to decode tokens from a well-trained AR model, the sampling error occurs in the predicted tokens which makes the decoder takes *noisy* and potentially out-of-distribution (OOD) latent patterns. This discrepancy necessitates that the latent space possess sufficient robustness to handle such latent perturbations, where reconstruction quality alone cannot capture. These observations motivate us to introduce robustness as an additional evaluation criterion and to design training strategies that explicitly enhance the tokenizer's ability to cope with noisy or perturbed latents.

Though robustness should be considered for tokenizer training as we illustrated above, it still cannot fully resolve the problem in sampling error of generator since robustness concerns only random perturbation in the latent space, whereas sampling errors are systematic and also involve perturbation from downstream models. Thus teaching the tokenizer how to interpret and reconstruct from generated tokens, rather than only clean tokens, is essential. However, this remains an open challenge due to the absence of explicit correspondence between each generated latent and its underlying ground truth image, making it more difficult to provide direct supervision for tokenizer post-training.

Building upon these insights, we further introduce a novel two-stage tokenizer training scheme to construct a robust latent space and decoder respectively. (1) During main-training (Fig. 1 (b)), we propose a novel plug-and-play discrete tokenizer training strategy that systematically integrates latent perturbations with an annealing schedule, gradually reducing perturbation intensity to stabilize training and promote robust latent space construction. (2) In post-training (Fig. 1 (c)), we design preservation ratio to control how much information from the target image is retained during generation, and use it to adapt the decoder to the distribution of latents produced by a well-trained generator. Extensive experiments conducted on state-of-the-art (SOTA) autoregressive frameworks (Sun et al., 2024; Yu et al., 2024b) across the ImageNet (Deng et al., 2009) generation benchmarks demonstrate the efficacy of our main-training. Moreover, across detailed experiments on off-the-shelf discrete and continuous tokenizers with their corresponding autoregressive and diffusion generative models, our post-training strategy also shows its broad applicability, consistently yielding promising improvements in generative quality. In particular, our method achieves 1.36 gFID with a ∼400M generator, establishing a new SOTA under this parameter budget.

Our contributions can be summarized as follows:

- We systematically analyze the discrepancy between reconstruction and generation in tokenizers, and provide the first comprehensive study on how robustness of the latent space impacts generative performance.

- We introduce RobusTok, a tokenizer trained using our novel two-stage training scheme including main-training for robust latent construction and post-training for generative latent alignment.

- We provide extensive experiments and ablation studies to validate the effectiveness of our latent perturbation and generalization ability of our post-training in autoregressive and diffusion generative models.

## 2 RELATED WORKS

**Image tokenizers.** Image tokenization has seen significant advancements across various image-related tasks. Traditionally, autoencoders (Hinton & Salakhutdinov, 2006; Vincent et al., 2008) have been employed to compress images into latent spaces for downstream applications such as generation and understanding. In generative tasks, VAEs (Van Den Oord et al., 2017; Razavi et al., 2019) learn to map images to probabilistic distributions; VQGAN (Esser et al., 2021; Razavi et al., 2019; Bachmann et al., 2025) and its subsequent variants (Lee et al., 2022a; Yu et al., 2023b; Mentzer et al., 2023; Zhu et al., 2024a; Takida et al., 2023; Huang et al., 2023; Zheng et al., 2022; Yu et al., 2023a; Weber et al., 2024; Yu et al., 2024a; Luo et al., 2024; Zhu et al., 2024b; Miwa et al., 2025) introduce discrete latent spaces to enhance compression and facilitate the application of autoregressive models (Vaswani et al., 2023; Dosovitskiy et al., 2021) to image generation tasks by converting images into sequences of discrete tokens. On the other hand, understanding tasks, such as CLIP (Radford et al., 2021), DINO (Oquab et al., 2023; Darcet et al., 2023; Zhu et al., 2024c) and MAE (He et al., 2022), rely heavily on LLM (Vaswani et al., 2023; Dosovitskiy et al., 2021) to tokenize images into semantic representations (Dong et al., 2023) where shown its promising performance in classification (Dosovitskiy et al., 2021), object detection, segmentation (Wang et al., 2021), and multi-modal application (Yang et al., 2024). In this paper, we provide a comprehensive analysis of image tokenizer in a view of perturbation robustness (Chen et al., 2024a; Li et al., 2024f; Xu et al.; Li et al., 2024g; 2023b;a).

**Autoregressive visual generation.** Autoregressive visual generation has shown remarkable success in generating high-quality images by modeling the distribution of pixels or latent codes in a sequential manner. Transformers (Vaswani et al., 2023), has demonstrated their strong capacity for capturing long-range dependencies and fine-grained details in image generation. Inspired by exploded development of language model (Shi et al., 2022; Mizrahi et al., 2024) such as GPT (Achiam et al., 2023), a series of works leverage tokenizers to convert images or visual information into discrete latent codes, enabling autoregressive or MLM modeling to generate image in raster-scan (Esser et al., 2021) or parallel (Pang et al., 2024a; Chang et al., 2022; Wang et al., 2024) order. Recently, autoregressive models continued to show their scalability power in larger datasets and multimodal tasks (He et al., 2024); models like LlamaGen (Sun et al., 2024) adapt current advanced LLM architectures for image generation. New directions such as VAR (Tian et al., 2024; Li et al., 2024e; Han et al., 2024; Ren et al., 2024; Qiu et al., 2024) and RAR (Yu et al., 2024b; Pang et al., 2024b) focus on fusing global information into the training of autoregressive model. MAR (Li et al., 2024b; Fan et al., 2024) and GIVIT (Tschannen et al., 2024) have shown the potential for continuous image generation. Through the development, various techniques continue to unify the large language model for generation and understanding (Wu et al., 2024; Tong et al., 2024).

## 3 PRELIMINARY - TOKENIZER ROBUSTNESS

**Vector Quantization (VQ).** Most AR models are based on discrete tokenizers with a quantized latent space. The tokenizer usually consists of an encoder, a quantizer, and a decoder. Although many quantization techniques for the quantizer were previously proposed (Mentzer et al., 2023; Yu et al., 2023b; Zhao et al., 2024), we focus on the VQ tokenizer (Esser et al., 2021) for its simplicity and natural compatibility with AR models in this paper.

Given an RGB image $I$, the encoder $\mathcal{E}$ first extracts a set of latent representations $Z \in \mathbb{R}^{H \times W \times C}$, where $H \times W$ denotes the spatial resolution of the latent tokens. VQ (Esser et al., 2021) aims to quantize continuous features into a set of discrete features $Z'$ with a minimum reconstruction error of the original data, ensuring that the quantized representation remains as close as possible to the original continuous ones. Specifically, it maps each continuous feature vector $z \in \mathbb{R}^C$ to a closest quantized codeword $\mathbf{e} \in \mathbb{R}^C$ from a learnable codebook $\mathcal{C} = \{e_k\}_{k=1}^K$ with in total $K$ codewords as:

$$z' = \arg\min_{e_k \in \mathcal{C}} \|z - e_k\|_2^2. \tag{1}$$

The decoder $\mathcal{D}$ then reconstructs the original input by taking the quantized $Z'$ as input.

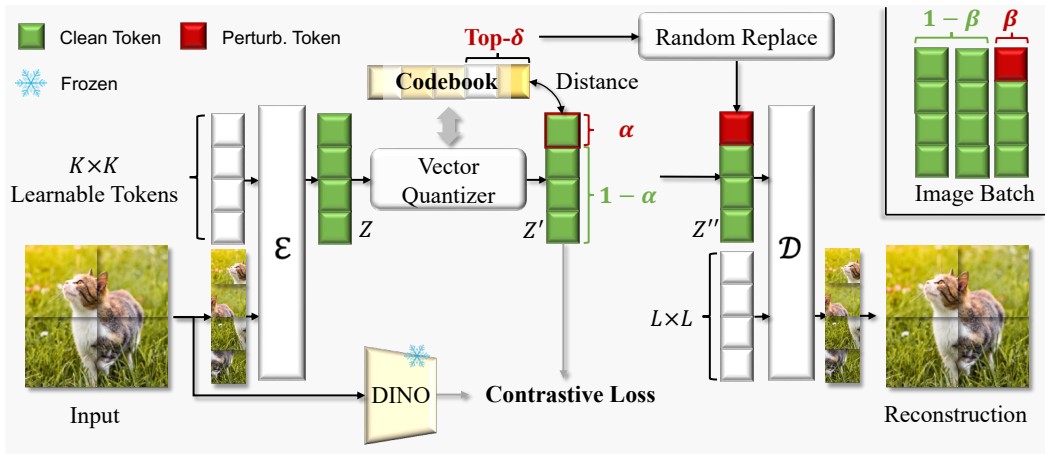

Figure 2: RobusTok overview. We adopt vision transformer as our encoder $\mathcal{E}$ and decoder $\mathcal{D}$. $\beta$ of data in one batch will process our Latent Perturbation, which will be randomly replaced by top-$\delta$ neighbor from codebook with probability $\alpha$. A frozen DINO encoder is utilized to supervise our latent space.

**Robustness.** Our method is motivated by mitigating the discrepancy between tokenizer training and inference schemes. As shown in Table 1, we demonstrate the input/output formulation of visual decoder upon the

| Ideal Scenario | Train Tokenizer | Eval AR/LLM |
|:---:|:---:|:---:|
| $\mathcal{D}(Z') = I$ | $\mathcal{D}(Z') = \hat{I}$ | $\mathcal{D}(Z' + \Delta) = \hat{I}'$ |

Table 1: Decoder analysis. $I$: ground-truth image. $\hat{I}$: predicted image. $\hat{I}'$: predicted image from noisy latent. $z'$: quantized latent feature. $\Delta$: sampling error. $\mathcal{D}$: decoder.

latent representations $Z'$. Ideally, the decoder $\mathcal{D}$ should take a clear latent $Z'$ and reconstruct the ground-truth image $I$ that aligns with the current tokenizer's training target. However, during the inference stage with a well-trained generative model, sampling error $\Delta$ always happens. This will change the usage of the decoder different from its training target, which significantly challenges the robustness of the visual decoder during inference as we expect $\mathcal{D}(Z' + \Delta)$ can still reconstruct the ground-truth $I$. To ease the discrepancy, RobusTok targets predicting ground-truth image from synthetic noisy latent $Z' + \Delta$ and real noisy latent $Z''$ from generative model during main-training and post-training respectively.

In addition, the robustness of the decoder can be measured by Lipschitz smoothness $Lip = \frac{\hat{I}' - \hat{I}}{\Delta} \approx \frac{\hat{I}' - I}{\Delta}$. Since the potential choice of $\Delta$ is constrained and the discrepancy between ground-truth $I$ and reconstructed images $\hat{I}'$ can be better reflected by the Fréchet Inception Distance (FID), we introduce perturbed FID (pFID) as a new metric to measure the robustness and reconstruction quality of tokenizers in our experiments where pFID = FID($\hat{I}', I$) (detailed in experiment section).

## 4 ROBUSTOK

RobusTok is a transformer-based image tokenizer shown in Fig. 2 with a two-stage training recipe. **Main-training**: constructing latent space with reconstruction target while involving synthetic perturbation to simulate sampling errors during generation. **Post-training**: generative finetuning tokenizer decoder to align with the well-trained generative model.

### 4.1 ARCHITECTURE

Following prior works (Li et al., 2024c;d; Yu et al., 2024d), RobusTok leverages Vision Transformer (ViT) (Dosovitskiy et al., 2021) as visual encoder and visual decoder. As shown in Fig. 2, we initialize a set of learnable tokens and use these tokens as the representation for image reconstruction and subsequent generation. Specifically, the input image is first patchified to $L \times L$ tokens, where $L$ represents the patch size, and concatenated with learnable tokens to serve as the input of the encoder. We apply vector quantization on the continuous token $Z$ obtained from the encoder $\mathcal{E}$. After that

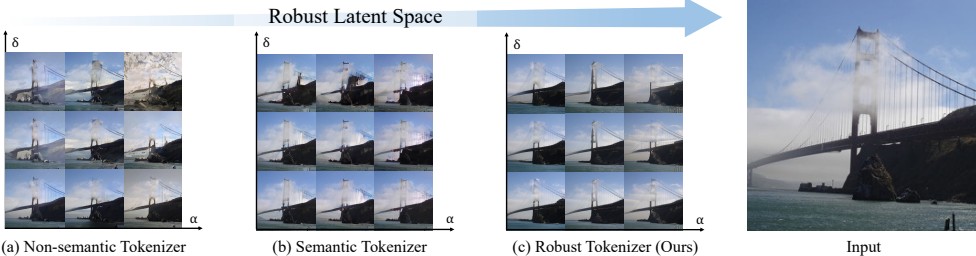

Figure 3: Visualization of (a) traditional tokenizer, (b) semantic tokenizer, and (c) our RobusTok in reconstruction task with Latent Perturbation. Non-semantic tokenizer leads to distorted reconstructions when perturbations are introduced while our method shows promising robustness to those perturbations.

a latent perturbation approach is applied to guide the latent space construction. Finally, the ViT decoder takes perturbed tokens $Z''$ and a new set of learnable tokens to reconstruct the image. Specifically, we incorporate a pretrained DINOv2 model (Oquab et al., 2023) to inject semantics, ensuring that the learned tokens retain meaningful visual semantics and structural coherence.

## 4.2 MAIN-TRAINING

In a discrete latent space, the latent space is constrained by a codebook. Thereby, sampling error happens in a form of token mismatch. As shown in Fig. 3, we define a set of operations to simulate the sampling error during tokenizer training.

**Perturbation rate.** An important metric to monitor the AR modeling process is the accuracy of predicted tokens. Likewise, we define a perturbation rate $\alpha$ to control the proportion of perturbed token within an image. Given the quantized feature $Z' \in \mathbb{R}^{H \times W \times C}$, we define $\alpha$ as:

$$\alpha = \frac{P}{H \times W}, \tag{2}$$

where $P$ denotes the perturbed token number. To simulate the sampling error, we can randomly perturb the quantized tokens from the tokenizer encoder.

**Perturbation proportion.** Within a batch of images, we apply the perturbation in a proportion $\beta$ of images and keep the remaining images unchanged. With $N_c$ clean images and $N_p$ perturbed images, the perturbation proportion is calculated as:

$$\beta = \frac{N_c}{N_c + N_p}. \tag{3}$$

**Perturbation strength.** We define a perturbation strength $\delta$ to quantify the perturbation level. Specifically, given a discrete token $z' = \mathbf{e}_k$ with a codebook $\mathcal{C}$, we calculate the set of top-$\delta$ nearest neighbors:

$$\mathcal{S}_\delta = \underset{\mathcal{S}_\delta \subset \mathcal{C}, |\mathcal{S}_\delta| = \delta}{\arg\min} \sum_{\mathbf{e}_n \in \mathcal{S}_\delta} \|\mathbf{e}_n - \mathbf{e}_k\|_2^2, \tag{4}$$

where $|\cdot|$ denotes the counting operation. We randomly replace the original token $\mathbf{e}_k$ with a $\mathbf{e}_\delta \in \mathcal{S}_\delta$ to perturb the latent, thereby modifying the latent representation to simulate sampling in AR with the top-k nucleus strategy.

**Plug-and-play perturbation.** During tokenizer training, we apply latent perturbation to enhance its robustness. We apply perturbation after semantic regularization (Li et al., 2024c) to preserve clear semantics in the discrete tokens to maximize the reconstruction capability. Within a batch of image, we randomly choose $\beta$ of them to add perturbation. To apply perturbation to each selected image, we randomly choose $\alpha \times H \times W$ tokens and then calculate the top-$\delta$ nearest neighbors to those tokens within the learned codebook. The final perturbation is applied by randomly replacing the original token with its top-$\delta$ nearest neighbor.

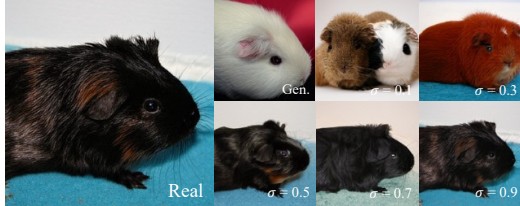

Figure 4: Generated images under different $\sigma$ for (left) autoregressive and (right) diffusion model.

### 4.3 POST-TRAINING

Following the training strategy described above, we obtain a more robust tokenizer. However, a gap still remains between the latents sampled from well-trained generator and those seen in tokenizer training, leading to generation degradation. To mitigate this issue, we introduce a lightweight post-training stage aimed at adapting the decoder to generated latents.

**Training scheme.** In this stage, we freeze the encoder and quantizer, and fine-tune only the decoder. To stabilize adversarial training, the pretrained discriminator is reused directly and optimization continues with the same loss combination. Concretely, our well-trained generator conditioned on each real image produces a generated image, which is re-encoded and paired with its corresponding real image, allowing the decoder to learn reconstruction from AR generated latents space.

**Preservation ratio.** To provide the decoder with meaningful guidance when learning from generated latents, each latent must be paired with its corresponding image. However, when relying solely on generated latents, the absence of the image pairs (generated latents v.s. ground truth generated image) directly blocks the decoder training. Inspired by the smooth transition induced by the teacher forcing (Sutton, 1988), we introduce the preservation ratio $\sigma$, which interpolates between real and generation by controlling how much information from the original image is retained in the generated latents. Specifically, in the latent space of the image tokenizer $Z' \in \mathbb{R}^{H \times W \times C}$, an autoregressive model generates an image by sequentially sampling tokens along the $H \times W$ grid. During each sampling step, we quantify the preservation ratio $\sigma$ to determine the ratio of tokens replaced by their ground-truth (real image) counterparts, formulated as

$$\sigma = \frac{N_{gt}}{H \times W} \tag{5}$$

where $N_{gt}$ denotes the number of tokens taken directly from the ground-truth latent. As shown in Fig. 4, this formulation allows $\sigma$ to smoothly control the trade-off between fully real and fully generated latents, thereby creating a connection between generated and real images and providing meaningful guidance for decoder training. Moreover, though teacher forcing is only applicable to autoregressive models, a similar mechanism can be realized in diffusion-based generation through SDEdit (Meng et al., 2021), a detailed explanation can be referenced to the Appendix.

## 5 EXPERIMENTS

### 5.1 EXPERIMENTAL SETTING

We experiment on ImageNet (Deng et al., 2009) 256×256 benchmark for both reconstruction and generation. We evaluate 11 open-sourced tokenizers across 4 codebook sizes. We follow their official implementation to pre-tokenize images and benchmark their generation performance using LlamaGen generators with default settings (Sun et al., 2024). For our RobusTok, we additionally leverage RAR (Yu et al., 2024b) as an additional generator to validate its wide applicability. In the post-training stage, we collect four representative tokenizers (including both discrete and continuous latent with autoregressive and diffusion model) to validate our method's effectiveness.

**Perturbed FID.** Except for typical Fréchet Inception Distance (FID) (Heusel et al., 2017), Inception Score (IS) (Salimans et al., 2016), Precision, and Recall for generator quality, we introduce pFID to assess tokenizer.

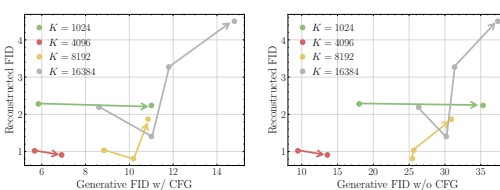 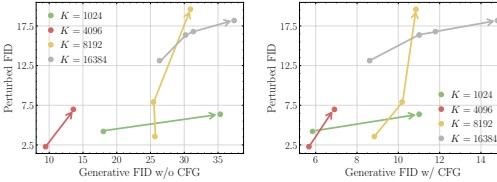

(a) **rFID** vs. gFID with and without CFG.  (b) **pFID** vs. gFID with and without CFG.

Figure 5: Comparison of rFID-gFID and pFID-gFID curves of different tokenizers under LlamaGen-B training setting. $K$ denotes codebook size. Each point represents a tokenizer in our benchmarking.

Compared to reconstruction FID (rFID) that merely captures the reconstruction quality of the tokenizer, pFID can reflect the robustness and the latent space from a tokenizer, and correlates with the sampling error and thus the performance of AR models.

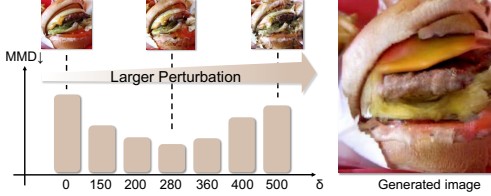

Figure 6: Maximum Mean Discrepency (MMD) between generated and $\alpha$-perturbed latent.

To calculate the pFID, we apply perturbation among all images, i.e., $\beta = 1$ for all the settings. In addition, we observe that, as shown in Fig. 6, choosing $\alpha \in [0.5, 0.9]$ and $\delta \in [200, 360]$ has a similar distribution with the generated latents. According to this, we define a set of perturbation rates $\alpha \in \{0.9, 0.8, 0.7, 0.6, 0.5\}$ and a set of perturbation strength $\delta \in \{200, 280, 360\}$ as the basis for computing pFID. For each of the $\alpha, \delta$ combinations, we generate perturbed reconstructions and compute the FID against the input images. The final pFID is obtained by averaging over all combinations. To ensure consistency across tokenizers, the perturbation strength $\delta$ is linearly scaled to match different codebook sizes.

Confirmed by our experiment in Section 5.2, our pFID is more correlated with the tokenizer's downstream generation performance compared with rFID.

**Implementation details.** During tokenizer training, we randomly select $\beta = 0.1$ of the total data to add perturbation. For these selected samples, we set $\alpha = 1.0$ and $\delta = 100$, and gradually anneal to half over the training. For the AR generator, we strictly follow the training recipes of LlamaGen (Sun et al., 2024) and RAR (Yu et al., 2024b). During post-training, we reduce the learning rate by half and reset the weight decay to zero. We use 16 NVIDIA A100 for main training, 8 NVIDIA RTX 4090 GPUs for post-training, and 8 NVIDIA RTX 4090 GPUs for all inference experiments.

## 5.2 MAIN EXPERIMENTS ANALYSIS

**General observations.** Before we go through and validate the core focus of this paper, we aim to conclude some generic observations from the benchmarking. The observations are summarized from the benchmarking results of LlamaGen-Base/Large.

- **Codebook size**: With similar reconstruction capability, the smaller the codebook size, the better the generation quality. We consider this property primarily results from the simple latent space are easier to capture during the AR modeling.
- **Semantics**: Semantic tokenizer typically demonstrates better capability for both reconstruction and generation. Semantic guidance provides a structural and clustering latent for better compression capability for reconstruction and robustness property for generation accordingly.
- **Reconstruction**: Reconstruction capability measured by traditional rFID does not align with the generation capability. This should be potentially resulted from the discrepancy between tokenizer training and inference, i.e., the latent space lacks robustness.

**Effectiveness of pFID.** To better compare the correlation among metrics, we visualize the rFID-gFID and pFID-gFID curves in Fig. 5 (a detailed value for each method & results for LlamaGen-Large generator are available in the Appendix). (a) When comparing rFID and gFID, we observe that there is no clear correlation between them, regardless of whether classifier-free guidance is used in generation or not. (b) Differently, pFID and gFID demonstrate a strong correlation within each

| Type | Method | Tokenizer | | Generator | | | | | | |
|---|---|---|---|---|---|---|---|---|---|---|
| | | rFID↓ | pFID↓ | gFID↓ | IS↑ | Pre↑ | Rec↑ | #Para | Leng. | Step |
| Diff. | ADM (Dhariwal & Nichol, 2021) | - | - | 10.94 | 101.0 | 0.69 | 0.63 | 554M | - | 1000 |
| | LDM-4 (Rombach et al., 2022) | - | - | 3.60 | 247.7 | - | - | 400M | - | 250 |
| | DiT-L/2 (Peebles & Xie, 2023) | 0.90 | - | 5.02 | 167.2 | 0.75 | 0.57 | 458M | - | 250 |
| | MAR-B (Li et al., 2024b) | 1.22 | - | 2.31 | 281.7 | 0.82 | 0.57 | 208M | - | 64 |
| NAR | MaskGIT (Chang et al., 2022) | 2.28 | 5.03 | 6.18 | 182.1 | 0.80 | 0.51 | 227M | 256 | 8 |
| | RCG (cond.) (Li et al., 2024a) | - | - | 3.49 | 215.5 | - | - | 502M | 256 | 250 |
| | TiTok-S-128(Yu et al., 2024d) | 1.52 | - | 1.94 | - | - | - | 177M | 128 | 64 |
| | MAGVIT-v2 (Yu et al., 2023b) | 0.90 | - | 1.78 | 319.4 | - | - | 307M | 256 | 64 |
| | MaskBit (Weber et al., 2024) | 1.51 | - | 1.65 | 341.8 | - | - | 305M | 256 | 64 |
| AR | VQGAN (Esser et al., 2021) | 7.94 | - | 18.65 | 80.4 | 0.78 | 0.26 | 227M | 256 | 256 |
| | RQ-Transformer (Lee et al., 2022b) | 1.83 | - | 15.72 | 86.8 | - | - | 480M | 1024 | 64 |
| | LlamaGen-L (Sun et al., 2024) | 2.19 | 13.12 | 3.80 | 248.3 | 0.83 | 0.52 | 343M | 256 | 256 |
| | VAR (Tian et al., 2024) | 0.90 | 17.46 | 3.30 | 274.4 | 0.84 | 0.51 | 310M | 680 | 10 |
| | ImageFolder (Li et al., 2024c) | 0.80 | 7.23 | 2.60 | 295.0 | 0.75 | 0.63 | 362M | 286 | 10 |
| | RAR (Yu et al., 2024b) | 2.28 | 5.03 | 1.70 | 299.5 | 0.81 | 0.60 | 461M | 256 | 256 |
| | RobusTok (Ours) | 1.02 | 2.28 | 1.60 | 305.8 | 0.78 | 0.65 | 461M | 256 | 256 |
| | + Tokenizer Post-Training | | | 1.36 | 300.2 | 0.77 | 0.66 | | | |

Table 2: System-level performance comparison on class-conditional ImageNet 256x256. ↑ and ↓ indicate that higher or lower values are better, respectively.

| Type | Method | Reconstruction | | | Generation w/o P.T. | | Generation w/ P.T. | |
|---|---|---|---|---|---|---|---|---|
| | | Latent | # Tokens ↓ | rFID ↓ | gFID ↓ | IS↑ | gFID ↓ | IS↑ |
| Diff. | MAETok (Chen et al., 2025a) | con. | 128 | 0.48 | 1.87 | 287.4 | 1.68 | 303.1 |
| AR | LlamaGen (Sun et al., 2024) | disc. | 256 | 2.19 | 3.80 | 243.8 | 3.51 | 241.2 |
| | GigaTok (Xiong et al., 2025) | disc. | 256 | 0.89 | 3.84 | 207.8 | 3.68 | 214.8 |
| | RobusTok-B | disc. | 256 | 1.02 | 1.83 | 298.3 | 1.60 | 288.0 |
| | RobusTok-L | disc. | 256 | 1.02 | 1.60 | 305.8 | 1.36 | 300.2 |

Table 3: System-level comparison on class-conditional ImageNet $256 \times 256$. "con." / "disc." denote continuous / discrete latent types. ↑ / ↓ indicate higher / lower is better. "P.T." represents our proposed tokenizer post-training strategy.

codebook size $K$. We separately compare results within each $K$ primarily because we add different perturbation strength $\delta$ according to $K$. With the new pFID, we can better access the tokenizer's performance without the time-consuming and resource-intensive training of subsequent generators.

**Systematic comparison.** As shown in Table 2, we compare our RobusTok with various state-of-the-art methods on the ImageNet $256 \times 256$ benchmark (Deng et al., 2009). In particular, RobusTok yields a notable improvement over prior approaches. Specifically, when applied on top of the RAR generator with the same training recipe, it achieves a 0.10 gFID gain. Moreover, with our simple tokenizer post-training strategy, our method attains new state-of-the-art generative performance among generators with fewer than 500M parameters, reaching 1.36 gFID. Even though our method relies on semantic supervision for better performance under smaller generators, latent perturbation still holds its performance under vanilla setting.

**Broad applicability of post-training.** To validate our post-training, we extend our method to four representative tokenizers covering both continuous and discrete tokenizers with their downstream diffusion and autoregressive models, respectively. As illustrated in Table 3, all methods achieve consistent improvements, demonstrating the broad applicability of our post-training strategy.

## 5.3 MORE ANALYSIS

**Robust latent space.** As shown in Fig. 7, we compare the latent space (i.e., codebook) with and without latent perturbation, we keep them training with same loss combination and exclude one without latent perturbation. We do the T-SNE calculation together and visualize their latent space separately. We colorize the latent tokens with their frequency of use during inference. When truncating tokens at differ-

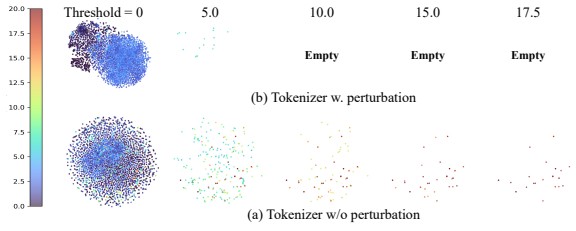

Figure 7: T-SNE visualization of latent space of tokenizer trained with and without latent perturbation. Colors and thresholds represent the frequency of tokens being used during inference without perturbation.

| ID | DINO supervision | Latent perturbation | rFID ↓ | PSNR ↑ | SSIM ↑ | gFID ↓ | IS ↑ |
|---|---|---|---|---|---|---|---|
| 1 | ✗ | ✗ | 1.77 | 20.4 | 0.66 | 4.28 | 217.7 |
| 2 | ✗ | ✓ | 1.63 | 20.5 | 0.67 | 3.89 | 222.8 |
| 3 | ✓ | ✗ | 1.68 | 20.5 | 0.66 | 3.78 | 228.3 |
| 4 | ✓ | ✓ | 1.60 | 20.6 | 0.68 | 3.64 | 231.4 |

Table 4: Ablation on latent perturbation without semantic supervision.

| $N$ | 0.2M | 0.5M | 1M |
|---|---|---|---|
| gFID | 1.60 | 1.60 | 1.59 |

(a) Training data.

| Epochs | 10 | 20 | 30 | 40 | 50 |
|---|---|---|---|---|---|
| gFID | 1.79 | 1.72 | 1.64 | 1.60 | 1.65 |

(b) Number of epochs.

| $\sigma$ | 0.5 | 0.6 | 0.7 | 0.8 | 0.9 |
|---|---|---|---|---|---|
| gFID | 1.98 | 1.83 | 1.79 | 1.79 | 1.80 |

(c) Preservation ratio.

Table 5: Design choices for RobusTok post-training.

ent usage count thresholds, we observe the space constructed with latent perturbation contains many reusable tokens, which acted as key tokens that can be easily modeled, while the remaining tokens serve as supportive tokens providing finer detailed information. In contrast, the latent space without latent perturbation distributes usage more uniformly across tokens.

**Latent perturbation without semantic supervision.** Following lines of recent works (Yao & Wang, 2025; Li et al., 2024c), semantic supervision has been widely been used for image tokenizer in reconstruction training. Our work, motivated by semantically structured latent space benefits generation, further isolate robustness attribute from semantic representation and conduct an additional ablation without semantic supervision, where we remove the DINO (Oquab et al., 2023; Li et al., 2024c; Yao & Wang, 2025) distillation loss and train the tokenizer only with standard VQ reconstruction and codebook losses, then apply our latent perturbation on top of this vanilla baseline. As shown in Table 4, latent perturbation still brings consistent improvements in generative metrics over the no-perturbation counterpart, while keeping reconstruction quality comparable. This indicates that the benefit of our latent perturbation is not solely driven by DINO-based semantic guidance, but provides an orthogonal robustness gain that also holds in the absence of external semantic supervision. Furthermore, after built upon tokenizer with semantic supervision, our proposed latent perturbation also shows its promosing performance.

**Perturbation selection & annealing strategy.** As shown in Table 6, we conduct an ablation to determine the optimal selection of perturbation hyperparameters. Our results indicate that using a large perturbation parameter, e.g., $\beta = 0.5$, degrades the model's reconstruction capability and adversely affects generative performance. Furthermore, training without annealing strategy leads to mode collapse and loss of generation diversity, whereas

| ID | Method | rFID↓ | pFID↓ | gFID↓ w./o. CFG | gFID↓ CFG |
|---|---|---|---|---|---|
| 1 | Baseline | 0.81 | 7.91 | 14.64 | 4.48 |
| 2 | + Codebook size 4096 | 0.91 | 6.98 | 7.91 | 4.13 |
| 3 | + Latent Perturbation $\beta = 0.5$ | 3.97 | 4.52 | 9.31 | 5.40 |
| 4 | + Latent Perturbation $\beta = 0.1$ | 1.58 | 3.61 | 4.60 | 3.93 |
| 5 | + Perturbation annealing to 0 | 0.97 | 4.89 | 5.32 | 1.97 |
| 6 | + Perturbation annealing to 0.5 | 1.02 | 2.28 | 4.62 | 1.85 |

Table 6: Ablation of RobusTok. gFID with classifier-free guidance (CFG) uses the constant schedule for LlamaGen and the linear schedule for RAR.

annealing to zero results in an overly deterministic tokenizer, diminishing the flexibility observed in Fig. 7. We find that annealing to half strikes a balance between robustness and adaptability, preserving essential latent properties while improving the quality of generated outputs. Other hyperparameters, e.g. perturbation rate and perturbation strength, exhibit stable performance over a reasonable range. We provide the corresponding sensitivity analysis in the appendix.

**Ablation for tokenizer post-training.** We conduct a systematic ablation study to investigate the impact of different design choices in tokenizer post-training. As shown in Table 5, increasing the amount of training data, or adjusting $\sigma$ does not necessarily lead to performance improvements. Instead, we find that the stability of training is the key factor for post-training effectiveness. This also explains why our RobusTok achieves a better performance improvement after post-training, as its decoder is inherently more robust due to the unique latent perturbation employed during pretraining. More experiment considering other tokenizers can be referred to the Appendix.

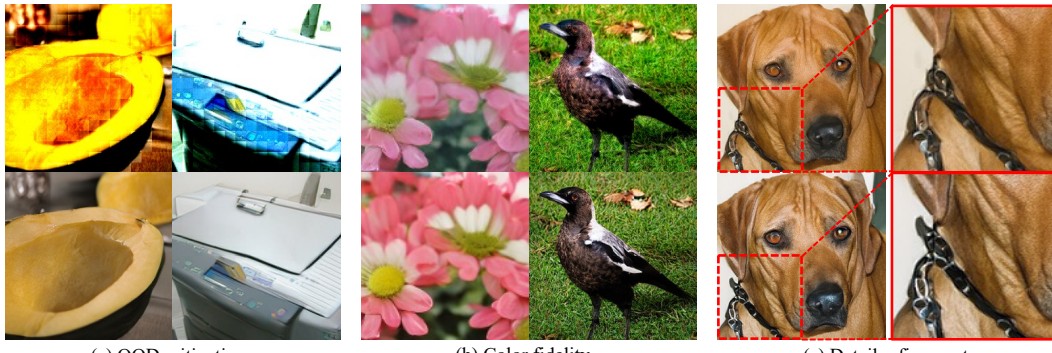

(a) OOD mitigation     (b) Color fidelity     (c) Detail refinement

Figure 8: Visualization of $256 \times 256$ image generation before (top) and after (bottom) post-training. Three improvements are observed: (a) OOD mitigation, (b) color fidelity, and (c) detail refinement.

**Qualitative results.** We visualize images generated before and after tokenizer post-training in Fig. 8. More generated result can be found in the Appendix.

## 6   CONCLUSION

In this paper, we explore the discrepancy between reconstruction and generation in tokenizer. To address this, we introduce a novel tokenizer training scheme including a plug-and-play latent-perturbation main-training to facilitate the construction of latent space, and a lightweight post-training stage to mitigate the degradation caused by distribution difference between generated and reconstructed latent space. Extensive experiments across both autoregressive and diffusion-based generators validate the effectiveness of our approach. We hope our research can shed light on the direction toward more efficient image representation and generation models.

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

# A APPENDIX

## A.1 CODEBOOK SIZE SELECTION

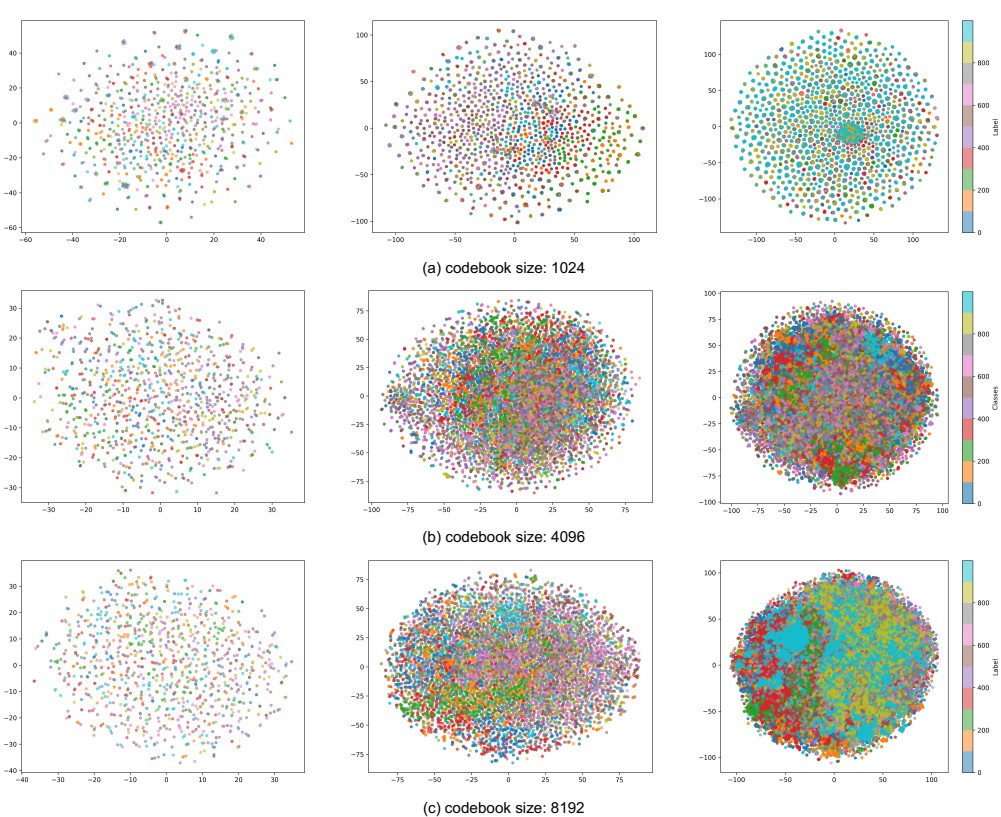

(a) codebook size: 1024

(b) codebook size: 4096

(c) codebook size: 8192

Figure 9: T-SNE visualization of latent space in baseline with varying codebook sizes setting: (a) 1024, (b) 4096, and (c) 8192. Each subfigure presents embeddings derived from (left) 1,000, (middle) 10,000, and (right) 50,000 samples from the ImageNet validation set. Compared to larger codebook sizes, XQGAN-1024 fails to maintain a well-structured latent space, leading to increased fragmentation and reduced robustness.

As described in ablation, we initialize our tokenizer with XQGAN-8192 Li et al. (2024d) as our baseline. Motivated by insights from Yu et al. (2024b); Weber et al. (2024) and our own benchmarking, we aim to reduce the codebook size for a more compact representation while preserving high

| Cluster Number | 512 | 1024 | 2048 | 4096 | 8192 | 16384 |
|---|---|---|---|---|---|---|
| SSE. | $2250_0$ | $1637_{-613}$ | $1253_{-384}$ | $928_{-325}$ | $611_{-317}$ | $473_{-138}$ |

Table 7: K-means clustering analysis of DINO features in ImageNet validation set. SSE. denotes as the Sum of Squared Error. The subscript values represent the difference in SSE. relative to the previous cluster number, indicating the reduction in error as the number of clusters increases.

reconstruction fidelity and generative quality. However, as shown in Fig. 9, the latent space of images in XQGAN-1024 appears highly fragmented, resulting in notable robustness discrepancies compared to tokenizers with larger codebooks, such as XQGAN-8192 and XQGAN-16384.

To better understand this, we analyze DINO features on ImageNet and apply k-means clustering to feature embeddings. As shown in Table 7, the results of the clustering of k-means, evaluated using the elbow method, indicate decreasing improvements in the Sum of Squared Errors (SSE) as the number of clusters increases beyond 4096. The reduction in SSE slows significantly at this point, suggesting that further increasing the number of clusters yields only marginal benefits. Based on this observation, we select $K = 4096$ as the codebook size for our tokenizer.

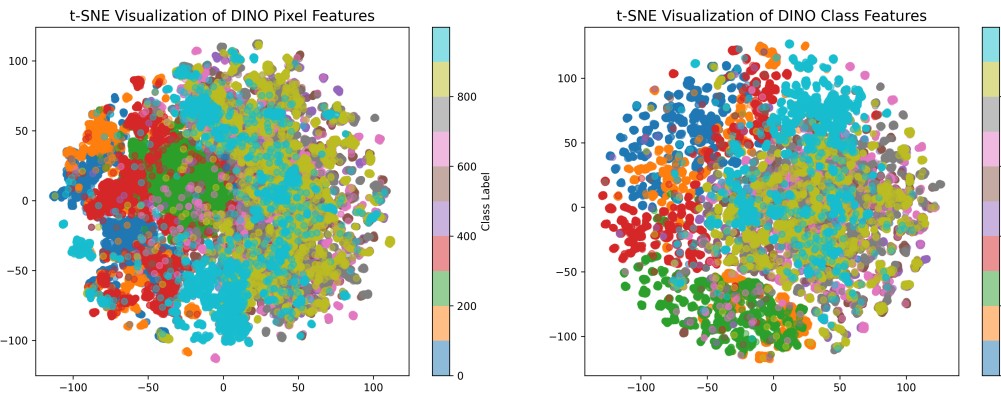

T-SNE visualization of DINO Pixel features.      T-SNE visualization of DINO Class features.

Figure 10: Visualization of DINO features in ImageNet Validation Set.

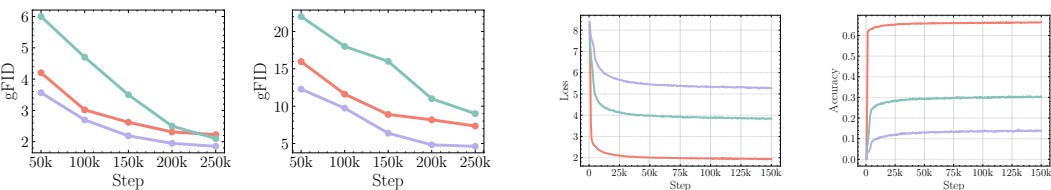

(b) Visualization of gFID trends for RAR, XQ-GAN, and Ours with (left) and without (right) CFG.

(a) RAR training loss (left) and accuracy (right) for None, Half, and Zero annealing strategies.

Figure 11: RAR training.

## A.2 LOSS FUNCTION.

The RobusTok is trained with composite losses including reconstruction loss $\mathcal{L}_{recon}$, vector quantization loss $\mathcal{L}_{VQ}$ Esser et al. (2021), adversarial loss $\mathcal{L}_{ad}$ Karras et al. (2019), Perceptual loss $\mathcal{L}_P$ Ledig et al. (2017), and semantic loss $\mathcal{L}_{clip}$ Li et al. (2024c):

$$\mathcal{L} = \lambda_{rec}\mathcal{L}_{rec} + \lambda_{VQ}\mathcal{L}_{VQ} + \lambda_{ad}\mathcal{L}_{ad} + \lambda_P\mathcal{L}_P + \lambda_{sem}\mathcal{L}_{sem}. \quad (6)$$

Specifically, the reconstruction loss measures the $L_2$ distance between the reconstructed image and the ground truth; vector quantization loss encourages the encoded features and its aligned codebook vectors; adversarial loss ensures that the generated images are indistinguishable from real ones; perceptual loss compares high-level feature representations to capture structural differences; and semantic loss performs semantic regularization between semantic tokens and the pre-trained DINOv2 Oquab et al. (2023) features. The only detail that requires special attention in our proposed latent perturbation is that the perturbation is added after the VQ/commitment loss, thereby emphasizing decoder robustness. Although we do not directly modify the encoder loss, the perturbation indirectly improves the encoder's representation quality by exposing the decoder to perturbed latents and enforcing stable reconstruction under noisy conditions.

**DINO supervision.** As shown in Fig. 10, we visualize the means of DINO pixel features and DINO class features. We observe that DINO class features exhibit a more structured representation compared to pixel-level features, which appear to be more scattered. Since the purpose of DINO features in our model is to provide supervision, the structured nature of class features makes them a more suitable choice to guide the learning process.

## A.3 RAR TRAINING

We follow the RAR training setting to validate the performance of our RobusTok. Specifically, as shown in Fig. 11, we evaluate RAR, XQGAN (our baseline), and our proposed RobusTok during

| Codebook Size | Method | Tokenizer Type | Tokenizer | | Generator | |
|---|---|---|---|---|---|---|
| | | | rFID↓ | pFID↓ | gFID↓ | gFID↓ (CFG) |
| 16384 | VQGAN-16384 (Esser et al., 2021) | Non-semantic | 4.50 | 18.18 | 37.39 | 14.80 |
| | LlamaGen (Sun et al., 2024) | Non-semantic | 2.19 | 13.12 | 26.34 | 8.61 |
| | IBQ-16384 (Shi et al., 2024) | Non-semantic | 1.41 | 16.35 | 30.19 | 11.01 |
| | VQGAN-LC (Zhu et al., 2024a) | Semantic* | 3.27 | 16.78 | 31.35 | 11.80 |
| 8192 | IBQ-8192 (Shi et al., 2024) | Non-semantic | 1.87 | 19.62 | 30.91 | 10.85 |
| | TiTok (Yu et al., 2024d) | Semantic | 1.03 | 3.55 | 25.66 | 8.84 |
| | XQGAN-8192 (Li et al., 2024d) | Semantic | **0.81** | 7.91 | 25.43 | 10.18 |
| 4096 | XQGAN-4096 (Li et al., 2024d) | Semantic | 0.91 | 6.98 | 13.58 | 6.91 |
| | RobusTok (Ours) | Semantic + Robust | 1.02 | **2.28** | **9.47** | **5.67** |
| 1024 | MaskGIT (Chang et al., 2022) | Non-Semantic | 2.28 | 4.20 | 18.02 | 5.85 |
| | IBQ-1024 (Shi et al., 2024) | Non-Semantic | 2.24 | 6.37 | 35.33 | 11.01 |

Table 8: Benchmark of tokenizers with the same LlamaGen-B generator. For fair comparison, the gFID with classifier-free guidance utilizes the same classifier value and schedule. All the tokenizers share the same $C \times 16 \times 16$ latent shape. We discuss the reason of choosing codebook size 4096 to train RobusTok in the ablation. More benchmarking results with larger generators are available in the appendix. * denotes semantics captured with linear projection. All metrics, i.e., rFID, pFID and gFID, are the smaller the better.

| Codebook Size | Method | Tokenizer Type | Tokenizer | | Generator | |
|---|---|---|---|---|---|---|
| | | | rFID↓ | pFID↓ | gFID↓ | gFID↓ (CFG) |
| 16384 | VQGAN-16384 (Esser et al., 2021) | Non-semantic | 4.50 | 18.18 | 20.89 | 6.23 |
| | LlamaGen (Sun et al., 2024) | Non-semantic | 2.19 | 13.12 | 8.61 | 4.40 |
| | IBQ-16384 (Shi et al., 2024) | Non-semantic | 1.41 | 16.35 | 21.57 | 5.53 |
| | VQGAN-LC (Zhu et al., 2024a) | Semantic* | 3.27 | 16.78 | 17.55 | 5.50 |
| 8192 | IBQ-8192 (Shi et al., 2024) | Non-semantic | 1.87 | 19.62 | 21.05 | 5.41 |
| | TiTok (Yu et al., 2024d) | Semantic | 1.03 | 3.55 | 14.51 | 4.47 |
| | XQGAN-8192 (Li et al., 2024d) | Semantic | **0.81** | 7.91 | 14.64 | 4.48 |
| 4096 | XQGAN-4096 (Li et al., 2024d) | Semantic | 0.91 | 6.98 | 7.90 | 4.13 |
| | RobusTok (Ours) | Semantic + Robust | 1.02 | **2.28** | **6.47** | **3.51** |
| 1024 | MaskGIT (Chang et al., 2022) | Non-Semantic | 2.28 | 4.20 | 12.37 | 3.60 |
| | IBQ-1024 (Shi et al., 2024) | Non-Semantic | 2.24 | 6.37 | 23.89 | 5.53 |

Table 9: Tokenizer benchmarking for LlamaGen-L. All metrics, i.e., rFID, pFID and gFID, are the smaller the better.

training. We observe that XQGAN achieves a faster convergence speed and better performance without CFG; however, its final performance, with a gFID of 2.22 under classifier-free guidance (CFG), remains suboptimal compared to RAR. Our RobusTok, inheriting the structural advantages of the semantic tokenizer while incorporating a robust latent space, not only achieves faster convergence but also outperforms both XQGAN and vanilla RAR in final generative quality, demonstrating its effectiveness in preserving semantic consistency and enhancing feature representation. This highlights a promising direction for designing more robust training schemes to further improve generative performance. Furthermore, the tokenizer without annealing exhibits strong convergence but compromises diversity, annealing to zero offers limited improvement over the baseline, while our annealing strategy provides a balance between generation diversity and quality.

### A.4 PFID RESULTS IN LLAMAGEN-L

As shown in Table 9 and Fig. 12, we further evaluate our perturbed FID (pFID) in the LlamaGen-L setting. Although the results contain some outliers, pFID still shows a stronger correlation with gFID than with rFID.

### A.5 SDEDIT FOR POST-TRAINING

For diffusion-based generation model, we employ SDEdit (Meng et al., 2021) to bridge reconstruction and generation. During sampling with a pre-trained diffusion model, we start from Gaussian

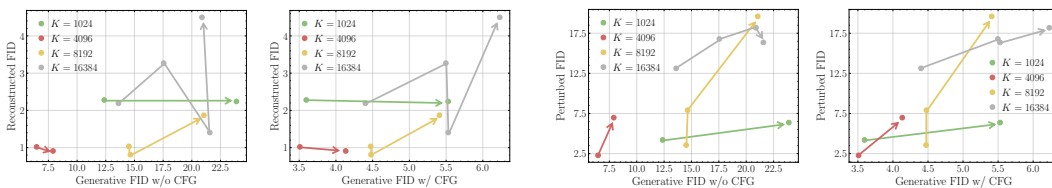

(a) **rFID** vs. gFID with and without CFG.

(b) **pFID** vs. gFID with and without CFG.

Figure 12: Comparison of reconstructed FID relation to generative FID with perturbed FID relation to generative FID. All generators follow LlamaGen-L training setting. K denotes as codebook size

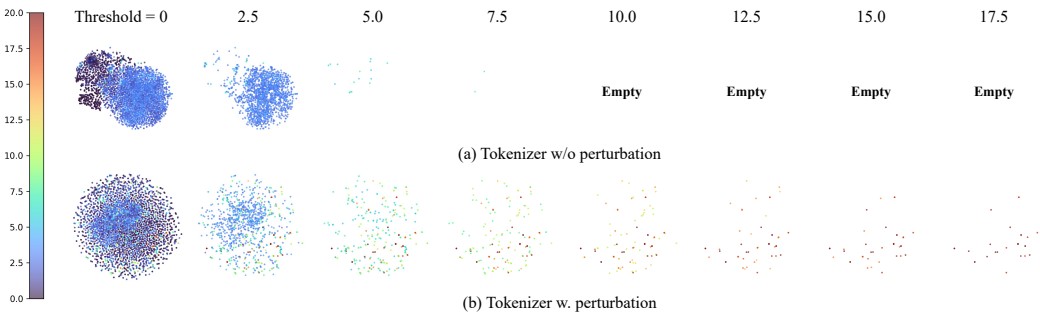

Figure 13: Detailed t-SNE visualization of latent space of tokenizer training with and without our proposed latent perturbation.

noise and gradually denoise it to obtain an image sample. In SDEdit, however, the process is initialized not from pure noise but from a real image corrupted by a controlled noise level. Following the preservation ratio $\sigma$ in the autoregressive case, we also define $\sigma$ for diffusion models as the fraction of the original latent space preserved after adding noise, formulated as

$$\sigma = 1 - \frac{t}{T} \tag{7}$$

where $t$ denotes the starting timestep and $T$ determines the total number of diffusion steps. By varying this noise level, we interpolate between reconstruction (small noise, more structure preserved) and generation (large noise, less structure preserved), making SDEdit a natural diffusion counterpart of our proposed method in autoregressive model.

### A.6 POST-TRAINING RESULTS

To find the optimal result for different methods under tokenizer post-training, we systematically design ablation studies for each tokenizer, as shown in Tables 10 and 11. We find that, although our RobusTok is relatively insensitive to the choice of $\sigma$, other tokenizers exhibit noticeable performance variation across different $\sigma$ values, highlighting the importance of proper preservation ratio selection for stable post-training.

| Epochs | baseline | $\sigma = 0.7$ | $\sigma = 0.8$ | $\sigma = 0.9$ | $\sigma = 0.95$ |
|--------|----------|----------------|----------------|----------------|-----------------|
| 10 | 3.80 | $4.60_{+0.80}$ | $4.06_{+0.26}$ | $3.77_{-0.03}$ | $3.51_{-0.29}$ |
| 20 | | $4.50_{+0.70}$ | $4.01_{+0.21}$ | $3.75_{-0.05}$ | $3.61_{-0.19}$ |

(a) **LlamaGen**

| Epochs | baseline | $\sigma = 0.6$ | $\sigma = 0.7$ | $\sigma = 0.8$ | $\sigma = 0.9$ |
|--------|----------|----------------|----------------|----------------|----------------|
| 10 | 3.84 | $4.15_{+0.31}$ | $3.88_{+0.04}$ | $3.68_{-0.16}$ | $3.75_{-0.09}$ |
| 20 | | $4.31_{+0.47}$ | $4.00_{+0.16}$ | $3.79_{-0.05}$ | $3.74_{-0.10}$ |

(b) **GigaTok**

Table 10: gFID in autoregressive models under different preservation ratio $\sigma$ and training epochs. For fair comparison, we randomly select 200,000 images from ImageNet training set. Baseline represents the original gFID using original checkpoints from (a) **LlamaGen** and (b) **GigaTok**.

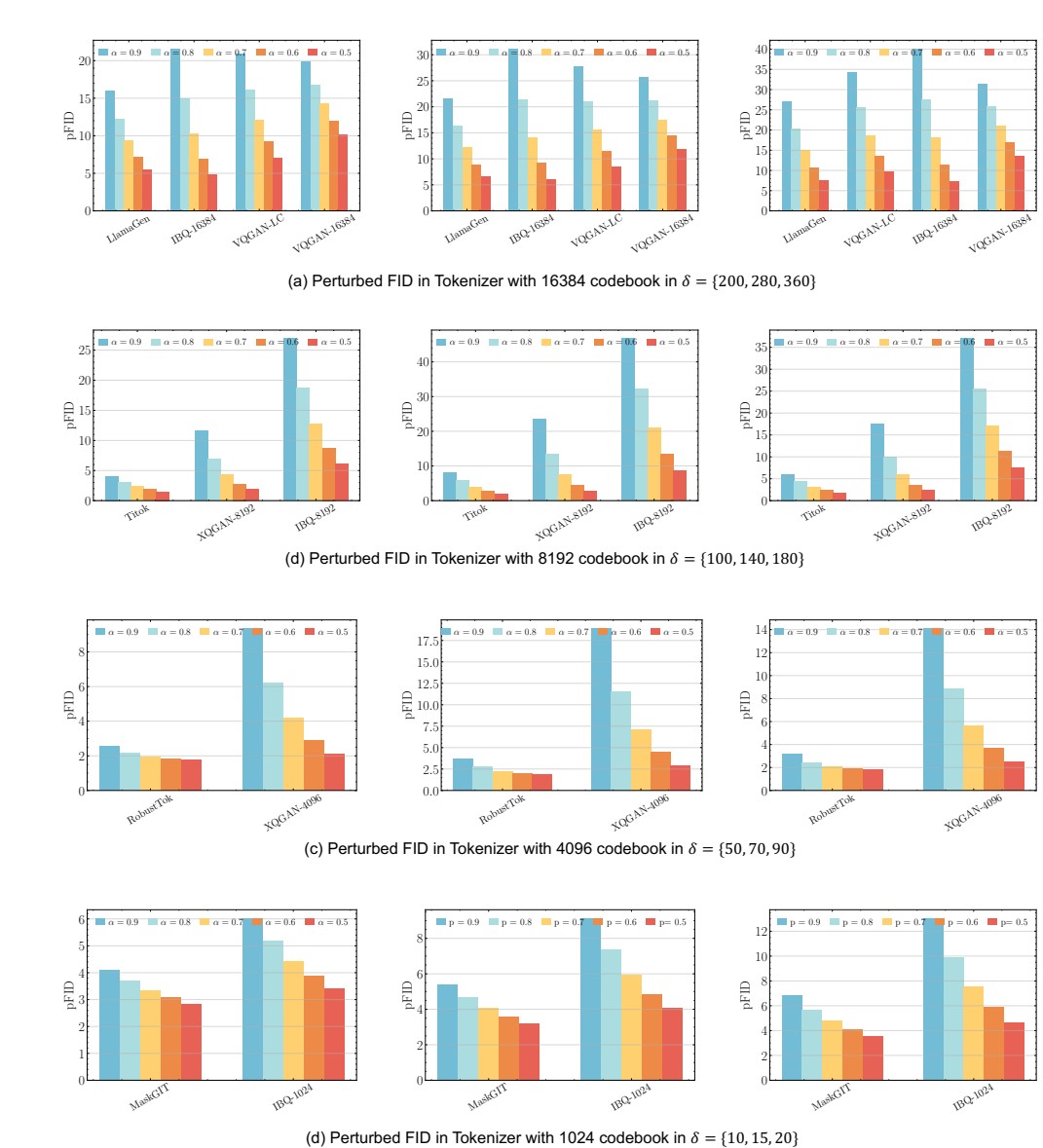

Figure 14: qualitative analysis of tokenizers in our latent perturbation.

| Epoch | baseline | $\sigma = 0.7$ | $\sigma = 0.8$ | $\sigma = 0.9$ |
|---|---|---|---|---|
| 10 | 1.87 | $1.82_{-0.05}$ | $1.76_{-0.11}$ | $1.68_{-0.19}$ |
| 20 | | $1.73_{-0.14}$ | $1.79_{-0.08}$ | $1.90_{+0.03}$ |

(a) **MAETok wo. finetuning**

| Epoch | baseline | $\sigma = 0.7$ | $\sigma = 0.8$ | $\sigma = 0.9$ |
|---|---|---|---|---|
| 10 | 1.74 | $2.00_{+0.26}$ | $1.81_{+0.07}$ | $1.70_{-0.04}$ |
| 20 | | $2.23_{+0.49}$ | $2.12_{+0.38}$ | $1.85_{+0.11}$ |

(b) **MAETok w. finetuning**

Table 11: gFID under different SDEdit strength and training epochs. For fair comparison, we randomly select 200,000 images from ImageNet training set. Baseline represents the original gFID using original checkpoints from MAETok (a) **without finetuning** and (b) **with latent noise finetuning**.

## A.7 LATENT PERTURBATION V.S. OTHER NOISES

To avoid potential misunderstanding, we aim to discuss the difference between our proposed latent perturbation and other noises used in generative models.

- **Latent perturbation**: Latent perturbation is a **random** noise manually added to the latent space based on the pattern we observed during the real sampling errors. Specifically, it is added in a cluster-based manner enlarging the decision boundary and zero-shot generalization during inference.

- **Diffusion noise**: Diffusion noise is a **scheduled** noise added to enable the reverse process using a diffusion sampler. It follows a pre-defined schedule to systematically disrupt the latent space.

- **Gaussian noise in VAE**: VAE's reparameterization employs a gaussian noise to decompose the mean value and randomness of the distribution to enable the gradient backpropagation.

- **Adversarial noise**: Adversarial noise in ML is widely used in adversarial and GAN-based training. It is an optimized perturbation designed to maximally **increase the loss or fool a classifier**. In contrast to our latent perturbation, adversarial noise is primarily designed to evaluate or improve robustness rather than to regularize tokenizers for better generative quality.

## A.8 VISUALIZATION

We demonstrate images generated by our approach as shown in Fig. 15. To further illustrate the effect of tokenizer post-training, we also present several failure cases before and after post-training for comparison. As shown in Fig. 16, Our tokenizer post-training is able to recover structural consistency, improve color fidelity, and sharpen local textures from original failed case.

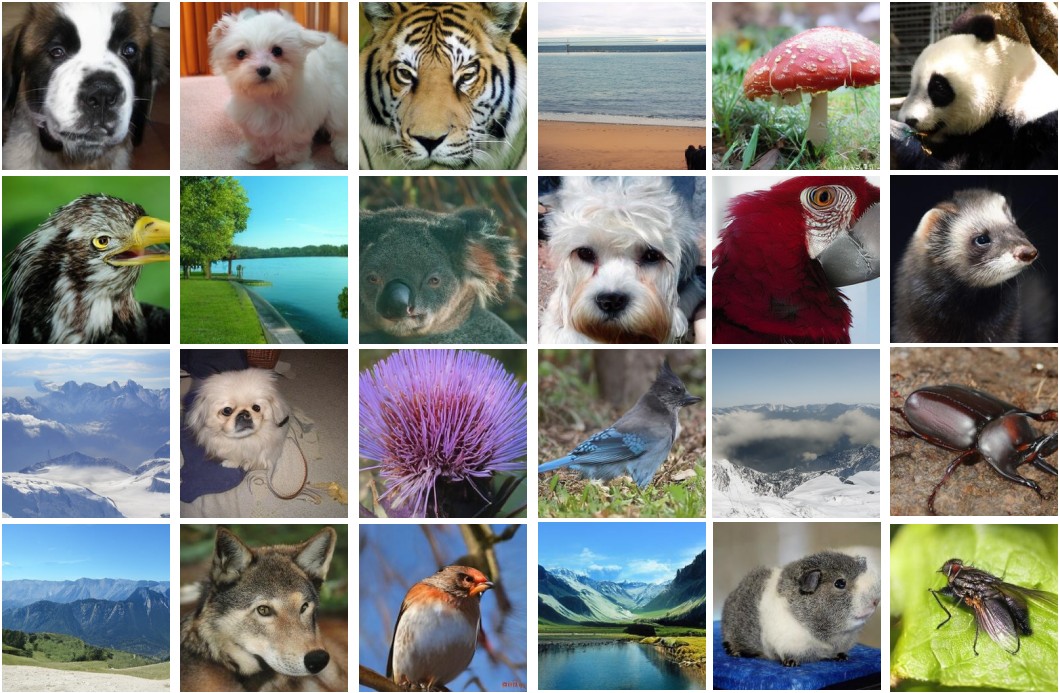

Figure 15: Visualization of 256 × 256 image within ImageNet class.

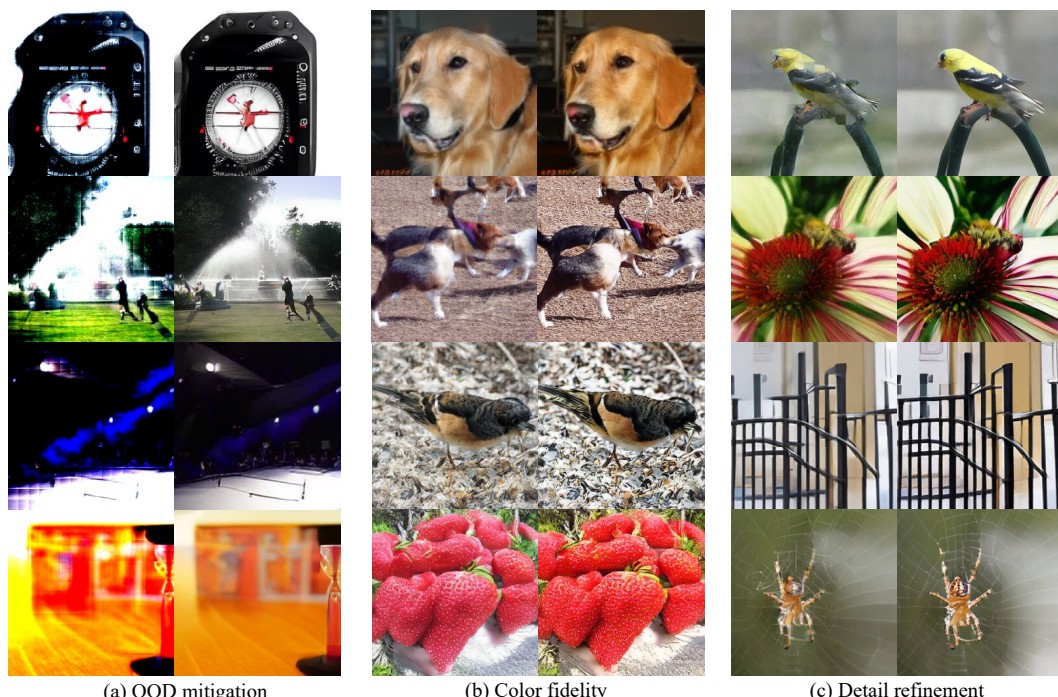

| (a) OOD mitigation | (b) Color fidelity | (c) Detail refinement |

Figure 16: More visualization for the improvement of tokenizer post-training in failed case.

| Encoder–Decoder Backbone | Total Params (M) | Latent Perturbation | rFID↓ | gFID↓ |
|---|---|---|---|---|
| ViT-S | 45M | ✗ | 2.37 | 9.80 |
| | | ✓ | 2.10 | 7.69 |
| ViT-B | 172M | ✗ | 1.77 | 5.66 |
| | | ✓ | 1.63 | 4.43 |

Table 12: Additional experiments about tokenizer size.

# B  ADDITIONAL EXPERIMENT

## B.1  ENCODER & DECODER DESIGN CHOICE

We demonstrate the ablation study to investigate the performance of our method under different encoder decoder parameters in Table 12. we select ViT-base as our final model based on its efficiency and performance.

## B.2  GENERALZIBILITY CROSS GENERATION

We further evaluate whether the proposed tokenizer post-training generalizes across different generators and under swapped generator–tokenizer pairings. As shown in Table 13, post-training consistently improves gFID for both RAR-B and RAR-L, regardless of which generator is used during post-training. Moreover, a tokenizer post-trained with RAR-B transfers well to RAR-L and vice versa, indicating that the learned robustness is not tightly to a specific generator.

## B.3  PERTURBATION RATE & PERTURBATION STRENGTH

For further deciding the optimal hyperparameter for perturbation rate and perturbation strength, we additionally do one ablation on determining the optimal values. As shown in Table 14, compared with significant performance in the annealing strategy and the proportion of perturbation in training, perturbation rate and strength does not fully fluctuate

| Generator | Tokenizer | gFID↓ | IS↑ |
|---|---|---|---|
| RAR-B | RobusTok (w/o P.T.) | 1.83 | 298.3 |
| | RobusTok (P.T. w/ RAR-B) | 1.60 | 288.0 |
| | RobusTok (P.T. w/ RAR-L) | 1.64 | 285.1 |
| RAR-L | RobusTok (w/o P.T.) | 1.60 | 305.8 |
| | RobusTok (P.T. w/ RAR-B) | 1.37 | 294.9 |
| | RobusTok (P.T. w/ RAR-L) | 1.36 | 300.2 |

Table 13: Comparison of generative performance for different generators and tokenizer post-training settings on ImageNet-256.

| Description | rFID↓ | gFID↓ | IS↑ |
|---|---|---|---|
| **Perturbation rate $\alpha$** | | | |
| $\alpha = 0.8$ | 1.59 | 3.92 | 218.9 |
| $\alpha = 0.9$ | 1.60 | 3.91 | 220.3 |
| $\alpha = 1.0$ | 1.60 | 3.89 | 222.8 |
| **Perturbation strength $\delta$** | | | |
| $\delta = 50$ | 1.55 | 3.95 | 219.4 |
| $\delta = 100$ | 1.60 | 3.89 | 222.8 |
| $\delta = 200$ | 1.63 | 3.85 | 224.6 |

Table 14: Ablation on perturbation rate $\alpha$ and perturbation strength $\delta$.

## C    LLM USAGE

We use large language models (LLMs) only for grammar checking and correction

