# OpenReview forum: "Image Tokenizer Needs Post-Training"
_ICLR.cc/2026/Conference — Submitted to ICLR 2026_

### Official Review · Reviewer_3GFk · 2025-10-24

**Soundness:** 2
**Presentation:** 2
**Contribution:** 2
**Rating:** 4
**Confidence:** 2

**Summary:**

This paper investigates the causes of performance discrepancies in discrete latent tokens and introduces a new tokenizer training framework composed of main-training and post-training stages. The main-training phase incorporates a latent perturbation strategy to mimic sampling noise and improve the robustness and efficiency of tokenizers. A training method and a new evaluation metric, namely pFID, are also proposed to better align tokenizer performance with generation quality. In the post-training phase, the tokenizer decoder is further optimized with respect to a trained generative model to reduce the gap between generated and reconstructed tokens.

**Strengths:**

[1]. The core idea is simple yet effective, and the authors provide solid experimental evidence to demonstrate its validity.
The comparisons with existing methods are fairly comprehensive, which helps position the contribution clearly within the current literature.

[2]. The paper presents extensive experiments and analyses that enhance the reader’s understanding of the proposed approach.
These results and discussions are valuable for future research in this area and may inspire subsequent work on improving image tokenizers and post-training strategies.

**Weaknesses:**

[1]. The design of the loss function appears overly complex, and there is no verification of parameter sensitivity for each hyperparameter. I suggest providing a table summarizing the optimal values of these parameters and explaining how those optimal values were determined.

[2]. It would be helpful to include experiments on the text-to-image task using JourneyDB. Without T2I results, it is difficult to assess the generalization ability of your method. Considering the complexity of your loss function, it may also be challenging for future researchers to effectively apply your approach in practice.

1. Sun K, Pan J, Ge Y, et al. Journeydb: A benchmark for generative image understanding[J]. Advances in neural information processing systems, 2023, 36: 49659-49678.

[3]. There is a typo in Table 4 (c): **Perservation ratio** should be corrected to **Preservation ratio.**

[4]. There are several issues in both the references and the OpenReview abstract, which suggest that this work may have been completed under significant time constraints.
4.1. The abstract on OpenReview contains the character sequence \ie, which appears to be an unprocessed LaTeX command.
4.2. The reference section includes numerous duplicated entries and formatting inconsistencies, indicating that the manuscript was not thoroughly proofread before submission. This raises concerns about the overall accuracy and attention to detail in the paper’s presentation.

*Alexey Dosovitskiy, Lucas Beyer, Alexander Kolesnikov, Dirk Weissenborn, Xiaohua Zhai, Thomas
 Unterthiner, Mostafa Dehghani, Matthias Minderer, Georg Heigold, Sylvain Gelly, et al. An
 image is worth 16x16 words: Transformers for image recognition at scale. arXiv preprint
 arXiv:2010.11929, 2020.*

*Alexey Dosovitskiy, Lucas Beyer, Alexander Kolesnikov, Dirk Weissenborn, Xiaohua Zhai, Thomas
 Unterthiner, Mostafa Dehghani, Matthias Minderer, Georg Heigold, Sylvain Gelly, Jakob Uszko
reit, and Neil Houlsby. An image is worth 16x16 words: Transformers for image recognition at
 scale, 2021. URL https://arxiv.org/abs/2010.11929.*

*Ali Razavi, Aaron Van den Oord, and Oriol Vinyals. Generating diverse high-fidelity images with
 vq-vae-2. Advances in neural information processing systems, 32, 2019a.*

*Ali Razavi, Aaron van den Oord, and Oriol Vinyals. Generating diverse high-fidelity images with
 vq-vae-2, 2019b. URL https://arxiv.org/abs/1906.00446.*

*J Ning, C Li, Z Zhang, Z Geng, Q Dai, K He, and H Hu. All in tokens: Unifying output space of
 visual tasks via soft token. arxiv 2023. arXiv preprint arXiv:2301.02229.*

*X Zhu, W Su, L Lu, B Li, X Wang, and J Dai. Deformable detr: Deformable transformers for
 end-to-end object detection. arxiv 2020. arXiv preprint arXiv:2010.04159, 2010.*

4.3. Some of the cited works appear to have low relevance to your study and could be removed.

[5] Figure 16 still contains many noticeable artifacts. For example, the text on the clock in the first row, first column is distorted; the bird’s tail in the first row, third column appears blurry; the human figure in the second row, first column fails to reconstruct properly; and the dog in the second row, second column is unclear. These examples demonstrate that, while your method shows certain improvements compared to the pre–post-training baseline, the overall visual quality remains subpar. This suggests that the effectiveness of your improvements is somewhat limited and raises questions about the actual contribution of the proposed pFID metric.

**Questions:**

[1]. Why not use adversarial noise? It would be interesting to explore how the model behaves when adversarial noise is applied. Would the proposed tokenizer or post-training strategy still maintain robustness under such perturbations?

[2]. How many GPU hours were required for training, and what kind of GPUs were used for inference? Please provide detailed computational requirements, including total GPU hours and the type or number of GPUs used. This information is important for assessing the reproducibility and scalability of your method.

[3]. What happens when using codes that are weakly correlated with the latent code? An analysis or ablation using latent codes that are less correlated with the original ones would help clarify the stability and robustness of the learned representation.

[4]. How does your method perform on datasets other than ImageNet? Evaluating on additional datasets would strengthen the claim of generalization and demonstrate that the proposed post-training method is not limited to ImageNet-specific settings.

---

> ### Author Response · Authors · 2025-11-20
>
> We thank the reviewer for the time and effort in reviewing our paper.
>
> ---
>
> **"loss function too complex? Optimal hyperparameter?"**
>
> We thank the reviewer for raising the concern about loss design complexity. However, our loss conceptually can be devidede into three parts: (1) standard VQ reconstruction and codebook losses, (2) an optional semantic distillation loss, (3) our proposed latent perturbation loss. we want to highlight that our loss is not complex but easy to understand and implement only in 20 lines code (select partion of data in one batch, given perturbation range and probability, randomly replace clean codes with perturbed codes). In practice, this adds only **four additional hyperparameters** on top of the standard VQ setup (perturbation probability, perturbation strength/range, and a simple schedule for when perturbation is active). We **explicitly ablate two of them in our experiments (Tab. 5)**, and for the remaining ones we observe that the performance is stable within a reasonable range as shown in Tab. A.
>
>
> | Setting                            | rFID ↓ | gFID ↓ | IS ↑ |
> |------------------------------------|--------|--------|------|
> | **Perturbation rate $\alpha$**     |        |        |      |
> | $\alpha = 0.8$                     |      1.59  |  3.92   | 218.9     |
> | $\alpha = 0.9$                    |      1.60  | 3.91    | 220.3     |
> | $\alpha = 1.0$                     |      1.60  | 3.89   |  222.8    |
> | **Perturbation strength $\delta$** |        |        |      |
> | $\delta=50$                        |  1.55  |  3.95  | 219.4  |
> | $\delta=100$                       |  1.60  |  2.89  | 222.8  |
> | $\delta=200$                       |  1.63  | 3.85    | 224.6 |
>
> Table A: additional experiment on ablating perturbation rate and perturbation strength.
>
> ---
>
> **"Experiments on the text-to-image task"**
>
> We thank the reviewer for suggesting additional text-to-image experiments. However, we would like to clarify that **text-to-image is not the primary scope of this paper**. Our goal here is to carefully study what kind of latent space and VQ-style tokenizer design is more “generator-friendly.” We believe that our current class-conditional ImageNet (c2i) experiments are sufficient to validate our main claims: (1) latent perturbation consistently helps the tokenizer achieve better generative performance, (2) pFID is more strongly correlated with gFID than rFID, and (3) tokenizer post-training can effectively reduce the training–inference gap between latents seen during tokenizer training and those produced during generation.
>
> ---
>
> **"Typo, unporcessed LaTeX, and duplicate reference"**
>
> We thank the reviewer for pointing out these issues in our manuscript. We will correct the typo in Table 4(c) (“Perservation ratio” → “Preservation ratio”), fix the unprocessed LaTeX command \ie in the OpenReview abstract, and thoroughly clean up the reference section to remove duplicated entries and formatting inconsistencies.
>
> ---
>
> **"Figure 16 artifact"**
>
> We would like to further clarify the purpose of Fig. 16. The goal of this figure is not to claim that our method completely eliminates all artifacts, but to **visually highlight how tokenizer post-training affects challenging ImageNet classes**. As shown in Fig. 16, before tokenizer post-training, the generator exhibits clear artifacts in hard categories. After applying our post-training procedure, these artifacts are substantially reduced and the global structure becomes more coherent, while fine details are better preserved.
>
> ---
>
> **"adversarial noise?"**
>
> In our paper, we use latent perturbation to **simulate the random errors that naturally occur during generator sampling**. Adversarial noise, in contrast, is constructed as a loss-maximizing perturbation, which does not reflect the sampling noise distribution. Therefore, adversarial noise does not fully match our target setting. We will add this clarification to the discussion of robustness in the revised version to make this distinction explicit.
>
> ---
>
> **"GPU for training & inference"**
>
> We use 16×A100 GPUs for the main training of the tokenizer and generator, and conduct tokenizer post-training on 8×RTX 4090 GPUs. All inference experiments are run on 8×RTX 4090 GPUs. tokenizer main training takes 2 days on 16 A100 for our final results and post training only take **less than 6 hours in 8xRTX 4090 GPUs**.
>
> ---
>
> **"Stronger perturbation?"**
>
> We would like to emphasize that our latent perturbation is **specifically designed to simulate the sampling errors that naturally occur during generator sampling**, rather than arbitrary hyperparameter. Pushing to much stronger perturbations would contradict the motivation behind pFID. In our preliminary checks, when the perturbation becomes too large (i.e., latents are effectively weakly correlated or nearly random), pFID quickly collapses and no longer reflects realistic generator behavior (see Fig. 6).

---

> > ### Comment · Reviewer_3GFk · 2025-11-21
> >
> > [1] Some of the cited works appear to be only weakly related to your study and could be removed.
> > **The papers by Xiang Li are cited 11 times, which seems excessive.** Could you clarify who Xiang Li is and explain why his work needs to be cited so frequently?
> >
> > [2] In my view, it should be feasible for you to conduct text-to-image experiments. Datasets such as JourneyDB are not very large, and from your paper I can see that your available devices are sufficient to support such experiments.
> >
> > Overall, the paper seems to have been completed under considerable time pressure, as there are many typos and incorrect citations. I hope you can address my concerns.

---

> > > ### Author Response · Authors · 2025-11-21
> > >
> > > We thank the reviewer for raising followup questions:
> > >
> > > ---
> > > 1. We consider all 11 papers are the theoretical / practical foundation of the proposed work. They are either about tokenizer or robustness. We would like to provide email address for Xiang Li: xli@x.ai (searched from his website). If the reviewer still have further questions about who is Xiang Li, we suggest the reviewer to schedule a phone call with him directly.
> > >
> > > This paper focuses on image tokenization, synthetic data and robustness. We list the relations of citation and this paper as follows:
> > > - **Slight corruption in pre-training data makes better diffusion model: latent noising and impact**: Diffusion model training with corrupted data augmentation.
> > > - **Masked autoencoders are effective tokenizers for diffusion models**: Image tokenization approach with masking corruption.
> > > - **Robust referring video object segmentation with cyclic structural consensus**: Vision language robustness with representation learning.
> > > - **Towards noise-tolerant speech-referring video object segmentation: Bridging speech and text**: Multimodal robustness with synthetic data and representation learning.
> > > - **Imagefolder: Autoregressive image generation with folded tokens**: State-of-the-art discrete image tokenizer with representation alignment.
> > > - **An open-source image tokenization framework for autoregressive generation**: State-of-the-art image tokenization codebase with multiple combination of advanced quantization and distillation.
> > > - **Controlvar: Exploring controllable visual autoregressive modeling**: State-of-the-art discrete image generation model.
> > > - **R2-bench: Benchmarking the robustness of referring perception models under perturbations**: Multimodal robustness with synthetic data augmentations.
> > > - **Qdformer: towards robust audiovisual segmentation in complex environments with quantization-based semantic decomposition.**: Audio-visual robustness with quantization and representation learning.
> > > - **Efficient autoregressive audio modeling via next-scale prediction**: State-of-the-art audio tokenizer with advanced quantization approach.
> > > - **Scalable benchmarking and robust learning for noise-free ego-motion and 3d reconstruction from noisy video**: Multimodal robustness with synthetic data and augmentations.
> > >
> > > The author hopes the reviewer to check all the citations again and we are more than welcome to provide further explanation.
> > >
> > > ---
> > >
> > > 2. We thank the reviewer for suggesting t2i setting with JourneyDB. In our reply, we have declared that our model is trained **less than 6 hours in 8xRTX 4090 GPUs** which is not the scale for t2i experiment. However, the authors are still willing to show results on two smaller internal datasets - **Open-NMYSL** and **Image-NDYYSL**. Hopefully, we will update and get the results before rebuttal ends.
> > >
> > > ---
> > >
> > > 3. We thank the reviewer for raising typo issues. We respectfully disagree with the reviewer's opinion that the paper is under time pressure. We hope the reviewer to consider our detailed and comprehensive experiments in the main paper and supplementary. If the reviewer can point to any typo and citation issues, we are more than welcome to address them.
> > >
> > > ---
> > >
> > > We are looking forward to your feedback.

---

> ### Author Response · Authors · 2025-11-20
>
> ---
>
> **"datasets other than ImageNet"**
>
> We agree that evaluating on additional datasets would further strengthen the generalization claim. Our proposed post-training only requires (i) inference access to the generator and (ii) full access to the parameters used in tokenizer training (including the discriminator needed for stable GAN-based post-training). However, current SOTA generator pipelines (e.g., FLUX [1] and Bagel [2]) are typically released without the corresponding discriminator, which prevents us from further post-training their tokenizers.
>
> ---
>
> Reference:
> * [1] FLUX.1 Kontext: Flow Matching for In-Context Image Generation and Editing in Latent Space
> * [2] Emerging Properties in Unified Multimodal Pretraining
>
> ---
>
> If you find our revise and response helpful, please consider raising the score for better support of our work.
>
> We are open to discuss more if any question still hold.
>
> Thanks!

---

### Official Review · Reviewer_ugEE · 2025-11-01

**Soundness:** 3
**Presentation:** 3
**Contribution:** 2
**Rating:** 2
**Confidence:** 4

**Summary:**

The paper studies the gap between reconstruction and generation in image tokenizers and introduces a two‑stage scheme called RobusTok. In the main training stage, a latent perturbation strategy is applied in discrete latent space to simulate sampling noise, yielding a tokenizer that is more robust to generation errors. A new metric, pFID, is proposed to better correlate tokenizer quality with downstream generation performance. In the post‑training stage, the tokenizer decoder is further optimized with respect to a well‑trained generator to mitigate the remaining distribution gap between reconstructed and generated tokens. With empirical experiments, it indicates effectiveness across both autoregressive and diffusion‑based generators and for off‑the‑shelf discrete and continuous tokenizers.

**Strengths:**

1. Clear analysis on the reconstruction–generation gap.
- In this paper, it analyzes the cause of the discrepancy in discrete latent space and introduces pFID, a metric designed to align tokenizer evaluation with downstream generative quality. The empirical study provides detailed evidence supporting the conjecture that robustness in latent space improves generation.

2. Method aligns the conjecture with practice.
- The proposed two-stage design, consisting of latent perturbation for robust latent construction and post-training for decoder alignment, closely follows the stated hypothesis and is validated through comprehensive experiments on both autoregressive and diffusion generators.

**Weaknesses:**

1. Loss design complexity and attribution.
- The loss formulation in Appendix A.2 makes tokenizer training appear somewhat complex. While the central conjecture is that simulating sampling error via latent perturbation reduces the reconstruction–generation gap, the perturbation is combined with several other loss terms, making the specific contribution of the perturbation harder to isolate.
- It remains unclear whether the observed gains persist without semantic regularization from an external vision encoder (e.g., DINO). A clear ablation that removes semantic regularization and isolates the effect of each loss component would strengthen the claims.


2. Unclear criteria for when post-training is beneficial.
- The paper motivates post-training as a means to address the residual gap between latents produced by a trained generator and those seen during tokenizer training. However, it remains ambiguous under what circumstances post-training should be applied. Since pFID is introduced as a diagnostic measure correlated with generation quality, clarifying whether a specific pFID threshold is used to trigger or terminate post-training would improve the claim.


3. Limited scope of experiments.
- The experiments are conducted primarily on the ImageNet 256×256 benchmark. It would strengthen the empirical validation to include additional experiments on ImageNet 512×512 or MS-COCO, demonstrating the generalization of the proposed method to higher resolutions and more diverse datasets.


4. Additional baselines for completeness.
- In Table 2 (class-conditional ImageNet 256×256), it would be beneficial to include references to FlexTok [1] and ResGen [2] within the related work discussion. Including these recent methods would strengthen the comparative context and highlight where the proposed approach stands relative to concurrent advances in tokenizer design.


References
- [1] FlexTok: Resampling Images into 1D Token Sequences of Flexible Length
- [2] Efficient Generative Modeling with Residual Vector Quantization-Based Tokens

**Questions:**

1. Figure 7 (t‑SNE) controls.
- In the t-SNE visualization comparing latent spaces with and without latent perturbation, were all other losses and regularizers, including semantic regularization from any external encoder, kept identical across conditions? Providing an explicit description of the controls or including an ablation without semantic regularization would make the attribution to latent perturbation more convincing.

2. Post‑training scheduling and stopping criteria.
- What metric and threshold determine when to start and when to stop post-training in practice? Is pFID used as a signal to initiate post-training, and is gFID or pFID employed as the stopping criterion? If pFID is intended as a practical diagnostic, it would be useful to know whether it consistently converges toward a target threshold during post-training across different models and datasets.

---

> ### Author Response · Authors · 2025-11-20
>
> We thank the reviewer for the time and effort in reviewing our paper.
>
> ---
>
> **"Loss design complexity and attribution."**
>
> We thank the reviewer for raising the concern about loss design complexity and attribution. The loss in Appendix A.2 can decomposes into: (i) standard VQ reconstruction and codebook losses, (ii) an optional semantic distillation term from an external vision encoder (DINO), and (iii) our proposed latent perturbation objective, which is the only component specifically introduced to simulate sampling errors and reduce the reconstruction–generation gap. The perturbation itself is implemented as a **small module of roughly 20 lines of Python code** (select partion of data in one batch, given perturbation range and probability, randomly replace clean codes with perturbed codes), making it easy to re-implement and plug into existing VQ-style tokenizers.
>
> Moreover, as shown in Tab. A, even without any DINO supervision, adding this latent perturbation consistently improves generative performance, indicating that its effect is not merely a byproduct of semantic distillation.
>
> | DINO supervision | Latent perturbation | rFID ↓| PSNR ↑ | LPIPS ↓ | gFID ↓ | IS ↑ |
> |------------------|---------------------|------:|-------:|-------:|--------:|-----:|
> | ✗                | ✗ (baseline)       | 1.77   | 20.4   | 0.66   | 4.28   | 217.7 |
> | ✗                | ✓                  | 1.63   | 20.5   | 0.67   | 3.89   | 222.8 |
> | ✓                | ✗          | 1.68   | 20.5   | 0.66   | 3.78   | 228.3 |
> | ✓                | ✓                  | 1.60   | 20.6   | 0.68   | 3.64   | 231.4 |
>
> Table A: Additional ablation experiment for latent perturbation. Our baseline here is vanilla VQGAN without semantic supervision.
>
> ---
>
> **"Unclear criteria for when post-training is beneficial."**
>
> We want to claim that our post-training stage is designed as a **generic refinement step** to reduce the distributional gap between latents seen during tokenizer training and latents produced by a trained generator. As shown in Table 3, we consistently observe that this post-training procedure improves generative quality **across all tokenizers we evaluate**, covering both discrete VQ-style tokenizers and continuous variants.
>
> ---
>
> **"Limited scope of experiments."**
>
> We respectfully disagree that our empirical validation is too limited to support the main claims. We deliberately conduct thorough experiments on ImageNet 256×256, which is the **standard benchmark for VQ-based tokenizers and autoregressive generators** and evaluate our method across multiple generators (LlamaGen [1] and RAR [2]), model scales, and extensive ablations. These results consistently show that our post-training and latent-robustness design does improve generative quality.
>
> ---
>
> **"Additional baselines for completeness."**
>
> We thank the reviewer for pointing out the concurrent works FlexTok [3] and ResGen [4]. Our Table 2, however, is explicitly designed as a controlled comparison under a ≤500M parameter budget (generator) for class-conditional ImageNet 256×256. FlexTok [3] and ResGen [4] pursue different design goals and operate under substantially different model/compute regimes, so they are not directly suitable for inclusion in Table 2’s efficiency-focused comparison. That said, we agree that they are highly relevant in terms of tokenizer design, and we will explicitly add them to the related work section and clarify our difference between them.
>
> ---
>
> **”Figure 7 Caption“**
>
> We thank the reviewer for this suggestion. For the t-SNE visualization in Fig. 7, all losses and regularizers are kept identical across conditions; the only difference between the two tokenizers is whether the proposed latent perturbation is enabled during training. We will revise our description (and the caption of Fig. 7) to further emphasize this controlled setting so that the attribution to latent perturbation is clearer.
>
> ---
>
> **"Post‑training scheduling and stopping criteria"**
>
> We want to emphasize that our post-training stage is designed as a **generic refinement method** for improving generative performance (see Tab. 3 in the paper). Since the cost of the proposed post-training is modest (only on **0.2M subset of ImageNet**), we simply run post-training until the generative performance converges. In our setup, a 40 epoches post-training run on this 0.2M subset takes only **less than 6 hours on 8xRTX 4090 GPUs**, which is negligible compared to the compute required to train the tokenizer & generator.
>
> ---
>
> Reference:
> * [1] Autoregressive Model Beats Diffusion: Llama for Scalable Image Generation
> * [2] Randomized Autoregressive Visual Generation
> * [3] FlexTok: Resampling Images into 1D Token Sequences of Flexible Length
> * [4] Efficient Generative Modeling with Residual Vector Quantization-Based Tokens
>
> ---
>
> If you find the above response help resolve your concerns, please consider raising score for better support of our work.
>
> Thanks!

---

> > ### Author Response · Authors · 2025-11-27
> >
> > Dear Reviewer,
> >
> > As the discussion period is approaching its conclusion, we would like to kindly check whether there are any remaining concerns or suggestions regarding our submission. If there is any further clarification or additional analysis that you would find helpful, we would be very happy to provide it. Your comments are extremely valuable in helping us strengthen this work.
> >
> > Thank you again for your time, effort, and thoughtful evaluation of our paper.

---

> ### Comment · Reviewer_ugEE · 2025-11-28
>
> Thank you for the detailed responses. The clarification on the loss components and the controlled setup for Figure 7 address my earlier concerns on attribution.
>
> I still have one remaining question regarding the practical role of pFID. In the paper, pFID is motivated as a metric that better reflects tokenizer robustness and correlates with downstream AR performance, which suggests it could serve as a useful diagnostic signal during tokenizer training or post-training. However, the current response mainly emphasizes that post-training is inexpensive and is simply run until “generative performance converges,” without clarifying how (or whether) pFID is used in this process.
>
> To better understand the intended use of pFID, could the authors briefly clarify:
>
> - Whether pFID is monitored during tokenizer training or post-training, and how it typically behaves over training; and
> - Whether the authors view pFID primarily as an evaluation metric, or also as a practical criterion (e.g., for deciding when to stop tokenizer training or when to trigger/terminate post-training).
>
> A short clarification on this point would help practitioners understand how to use pFID beyond retrospective analysis.

---

> > ### Author Response · Authors · 2025-11-28
> >
> > Thank you for the helpful follow-up question. We are happy to clarify how pFID is used in practice.
> >
> > ---
> >
> > **"use of pFID"**
> >
> > In our experiments, we use pFID as an additional **evaluation metric** on imagenet validation set. We **only use pFID in tokenizer main training** for evaluating whether the current tokenizer is robust enough to tolerate latent perturbations happened in generation. We **do not use our pFID as beginning or stopping criterion in post training** because post training is aligning the error happened in generator with real image so that even wrong prediction in generator can also get a reasonable image. We want to highlight that the main contribution in our paper is **Latent perturbation in main training & tokenizer post training** for a sota performance in roughly small generator budget.
> >
> > ---
> >
> > We are looking forward to your additional feedback.

---

### Official Review · Reviewer_3zmd · 2025-11-01

**Soundness:** 3
**Presentation:** 2
**Contribution:** 2
**Rating:** 4
**Confidence:** 5

**Summary:**

The authors argue that conventional tokenizers, optimized solely for reconstruction fidelity, lack the necessary robustness to handle the noisy and out-of-distribution latent codes produced during the generative inference stage. To solve this, they propose a two-stage training scheme called "RobusTok". The first stage introduces a latent perturbation strategy to build a robust latent space by simulating sampling noises. The second stage, post-training, further refines the tokenizer's decoder to align it with the specific distribution of latents generated by a pre-trained generative model. The paper presents a new evaluation metric, pFID, to correlate tokenizer performance with generation quality. Through extensive experiments on autoregressive models, the authors demonstrate that their method significantly improves generation quality.

**Strengths:**

- The paper provides a novel analysis of the discrepancy between reconstruction and generation performance in token-based generative models. It argues that tokenizer robustness, not just reconstruction fidelity, is a critical factor for high-quality image synthesis.
- The proposed two-stage training scheme presents a practical and effective solution.

**Weaknesses:**

- The proposed method of injecting noise into the latent space to achieve robustness bears a strong resemblance to VAE. To properly contextualize this contribution, the authors should provide a direct comparison with VAVAE [1], another relevant work that also leverages VAE and DINO features.

- The paper's ablation studies should be expanded to include experiments conducted *without* the DINO distillation loss. First, to disentangle the performance gains attributable to the proposed latent perturbation from those provided by DINO; and second, to validate whether the noise injection strategy remains effective in the absence of DINO semantic guidance. Furthermore, the motivation for incorporating DINO distillation is not explicitly justified and appears to be treated as a default choice, whereas its role and necessity should be clearly articulated.

- As shown in Tables 5 and 7, the proposed latent perturbation strategy, while boosting generative performance, consistently degrades reconstruction quality. This introduces a critical trade-off between generation and reconstruction that the authors should discuss in detail. A thorough analysis of this trade-off is needed to understand its implications and to clarify whether this degradation is a fundamental compromise or a side effect that could be mitigated.

[1] Reconstruction vs. Generation: Taming Optimization Dilemma in Latent Diffusion Models

**Questions:**

See weakness

---

> ### Author Response · Authors · 2025-11-20
>
> We thank the reviewer for the time and effort in reviewing our paper.
>
> ---
>
> **"Comparison with VAVAE [1]"**
>
> We appreciate the reviewer’s pointer to VAE-style modeling and VAVAE [1]. Our method is indeed motivated by the empirical observation that semantic distillation can help generative models. In this work, **we go one step further and hypothesize that latent robustness is a key reason behind such improvements**; this directly motivates our proposed pFID metric as well as the latent perturbation scheme used in tokenizer training. We will cite VAVAE [1] in the related work and we already include a discussion on how our VQ-based latent perturbation relates to, and differs from, VAE-style noise injection in supplementary A.7.
>
> ---
>
> **"experiments without DINO."**
>
> Our use of a DINO distillation loss follows line of recent works [1] showing that distilling strong self-supervised features into the tokenizer improves semantic structure of latents and stabilizes training. In our setting, DINO distillation is used to ensure that the learned codebook preserves high-level semantics, while the proposed latent perturbation specifically targets robustness of individual codes and reduces decoder artifacts. For further investigate our latent perturbation alone, we add ablations trained with and without the DINO distillation loss. As shown in Tab. A, our latent perturbation achieves improvement in generative metrics under both setting.
>
> | DINO supervision | Latent perturbation | rFID ↓| PSNR ↑ | LPIPS ↓ | gFID ↓ | IS ↑ |
> |------------------|---------------------|------:|-------:|-------:|--------:|-----:|
> | ✗                | ✗ (baseline)       | 1.77   | 20.4   | 0.66   | 4.28   | 217.7 |
> | ✗                | ✓                  | 1.63   | 20.5   | 0.67   | 3.89   | 222.8 |
> | ✓                | ✗          | 1.68   | 20.5   | 0.66   | 3.78   | 228.3 |
> | ✓                | ✓                  | 1.60   | 20.6   | 0.68   | 3.64   | 231.4 |
>
> Table A: Additional ablation experiment for latent perturbation. Our baseline here is vanilla VQGAN without semantic supervision.
>
> ---
>
> **"Trade-off between generation and reconstruction."**
>
> We thank the reviewer for highlighting the trade-off between generative and reconstruction performance. Indeed, our latent perturbation strategy improves generative metrics while slightly degrading reconstruction metrics. This behavior further supports our design of pFID as a more generation-aligned objective, as it tracks improvements in sample quality even when pixel-level reconstruction slightly drops (shwon in Table 3 and 5). Furthermore, under our final setting with latent perturbation + perturbation annealing, the tokenizer achieves reconstruction metrics comparable to the baseline while still enjoying a clear FID gain (as shown in Tab. A). This suggests that the trade-off is not a fundamental limitation, but rather a controllable effect that can be largely mitigated by a proper perturbation schedule, allowing us to obtain a more “generator-friendly” latent space without sacrificing reconstruction quality.
>
> Reference:
> * [1] Reconstruction vs. Generation: Taming Optimization Dilemma in Latent Diffusion Models
>
> ---
>
> If you find our revise and response helpful, please consider raising the score for better support of our work.
> We are open to discuss more if any question still hold.

---

> > ### Author Response · Authors · 2025-11-27
> >
> > Dear Reviewer,
> >
> > As the discussion period is drawing to a close, we would like to confirm that we have addressed all of your concerns. If you have any further comments or suggestions, we would be grateful to hear them. Your feedback is extremely valuable for improving our work.
> >
> > Thank you again for your time and careful review.

---

### Official Review · Reviewer_Vx6s · 2025-11-01

**Soundness:** 3
**Presentation:** 3
**Contribution:** 3
**Rating:** 6
**Confidence:** 4

**Summary:**

The authors address the mismatch between clean encoder latents and generated tokens produced by a pretrained generator.
To mitigate this discrepancy, they first train the tokenizer with random latent perturbations, improving decoder robustness to noise.
In a second stage, they fine-tune the decoder on the generator’s actual reconstruction errors to further align it with the generator’s distribution.

**Strengths:**

- Achieves strong FID despite using a relatively small model.
- Clearly demonstrates the weakness of training tokenizers independently from generators.
- Introduces a simple yet effective strategy to simulate generator errors through post-training.

**Weaknesses:**

- Requires an additional ~50 epochs of training — could this process be done jointly with generator training?
- The generator’s parameter count is small, but encoder–decoder size and efficiency are not discussed.
- While rFID improves, comparisons on PSNR or SSIM would strengthen the evaluation.

**Questions:**

- Is each tokenizer specific to a given generator, or can it generalize across generators?
- FID is strong, but how does it perform on the validation set? Could the model be overfitting? What about visualization on more challenging classes where other generator usually fails?
- Could the authors elaborate on the role and impact of the DINO contrastive loss?
- pFID correlates more closely with gFID than rFID. why does this relationship hold?

---

> ### Author Response · Authors · 2025-11-20
>
> We thank the reviewer for the time and effort in reviewing our paper.
>
> ---
>
> **"Training Time for Post Training? Possibility for joinly training?"**
>
> We want to highlight that our proposed post-training stage is actually very lightweight. We only train on 0.2M ImageNet, and the entire 40 epoch training takes **less than 6 hours on 8xRTX 4090 GPUs** in our setup. This cost is negligible compared to the compute typically required to train the generator itself. In this paper we deliberately keep it as a separate stage to cleanly isolate its effect and to make the method plug-and-play for existing generators, and exploring joint training is another direction that we leave for future works.
>
> ---
>
> **"encoder–decoder size and efficiency."**
>
> We would like to clarify that, the objective of our paper is to investigate what kind of latent space and VQVAE design is more “generator-friendly”, rather than to aggressively scale the generator or encoder–decoder. We also include a preliminary ablation on the encoder–decoder size. We select ViT-B as our encoder and decoder because it offers a good trade-off between capacity and efficiency. As shown in Tab. A, our gains are complementary to backbone scaling.
>
> | Encoder–Decoder Backbone | Total Params (M) | Latent Perturbation | rFID ↓ | gFID ↓ |
> |--------------------------:|----------------:|--------------------:|-------:|-------:|
> | ViT-S                    | 45M              |        ✗            |  2.37  | 9.80   |
> |                          |                  |        ✓            |  2.10  | 7.69   |
> | ViT-B                    | 172M             |        ✗            |  1.77  | 5.66  |
> |                          |                  |        ✓            |  1.63  | 4.43  |
>
> Table A: Additional experiments about tokenizer size.
>
> ---
>
> **"More evaluation metrics"**
>
> Although rFID improves, this is not the central objective of our study. Our primary goal is to demonstrate that latent perturbation and subsequent post-training help close the training–inference gap in the tokenizer, rather than to optimize low-level reconstruction metrics. For completeness, however, we also report PSNR and SSIM in Table C for both the baseline and the baseline with latent perturbation, and both metrics show consistent improvements, further supporting the effectiveness of our approach.
>
> ---
>
> **"generalize across generators?"**
>
> Our design is not tied to a specific generator. The proposed tokenizer is generator-agnostic and can be plugged into different generator backbones as long as they operate on the same latent space. We provide additional experiments on tokenizer post-training with different generators and swapped generator–tokenizer pairings. As shown in Tab. B, for each generator (RAR-B and RAR-L) we compare RobusTok without post-training and with post-training using either RAR-B or RAR-L, all of them achieve improvements.
>
> | Generator        | Tokenizer Used            | FID ↓ | IS ↑  |
> |------------------|---------------------------|------:|------:|
> | **Baseline**     |                           | 1.83  | 298.3 |
> | RAR-B            | RAR-B tokenizer (orig.)   | 1.60  | 288.0 |
> | RAR-B            | RAR-L tokenizer (swap)    | 1.64  | 285.1 |
> | **Baseline**     |                           | 1.60  | 305.8 |
> | RAR-L            | RAR-L tokenizer (orig.)   | 1.36  | 300.2 |
> | RAR-L            | RAR-B tokenizer (swap)    | 1.37  | 294.9 |
>
> Table B: Experiment on tokenizer post-training with different generator.
>
> ---
>
> **"Post Training visualization"**
>
> we agree that qualitative behavior there is important. We already include such visualizations in **Fig. 8 and Fig. 16**, and find our post-training systemically improve dromastically in ImageNet hard classes. As shown in these figure, we can conclude that our tokenizer post training improve generative performance especially structural consistency, color fidelity, and local textures from original failed case.
>
> ---
>
> **"Impact of the DINO loss?"**
>
> We conduct another ablation to validate the performance directly lead by our latent perturbation. As shown in Tab. C, we start with our baseline without dino supervision and further train it with our proposed latent perturbation. We find that our proposed latent perturbation achieve a good performance with and without semantic supervision.
>
> | DINO supervision | Latent perturbation | rFID ↓| PSNR ↑ | LPIPS ↓ | gFID ↓ | IS ↑ |
> |------------------|--------------------|-------:|-------:|-------:|--------:|-----:|
> | ✗                | ✗ (baseline)       | 1.77   | 20.4   | 0.66   | 5.66   | 163.01 |
> | ✗                | ✓                  | 1.63   | 20.5   | 0.67   | 4.24    | 202.33     |
> | ✓                | ✗          | 1.68   | 20.5   | 0.66   | 3.78   | 228.3 |
> | ✓                | ✓                  | 1.60   | 20.6   | 0.68   | 3.64   | 231.4 |
>
> Table C: Additional ablation experiment for latent perturbation. Our baseline here is vanilla VQGAN without semantic supervision.

---

> > ### Author Response · Authors · 2025-11-20
> >
> > ---
> >
> > **"Why pFID?"**
> >
> > We observe that pFID correlates more closely with gFID than rFID because rFID tests the decoder on clean tokenizer codes extracted from real image. This matches the training distribution of the decoder but does not reflect the actual codes seens at generator inference. pFID, in contrast, measures the full generation pipeline with synthesizing latent perturbation happening in generator inference.
> >
> > ---
> >
> > We hope the above response can resolve your concern. If you find our revise and response helpful, please consider raising the score to better support of our work. Also please let us know if there is any more question.

---

> > > ### Author Response · Authors · 2025-11-27
> > >
> > > Dear Reviewer,
> > >
> > > I hope this message finds you well. As the discussion period is nearing its end. I want to ensure we have addressed all your concerns. If there are any additional points or feedback you'd like us to consider, please let us know. You insights are invaluable for us to improve our work.
> > >
> > > Thank you for your time and effort in reviewing our paper.

---

### Meta-Review · Area_Chair_ALAZ · 2026-01-07

**Summary:**

The submission proposes a two-stage tokenizer training framework to bridge the gap between reconstruction and generation, but reviewers expressed concerns regarding its unclear attribution of gains and lack of empirical depth. Specifically, the necessity of the DINO loss remains unverified through proper ablations, and the evaluation is limited by missing recent baselines and a noticeable trade-off in reconstruction fidelity.

**Reviewer Concerns:**

Despite the authors' clarifications and new experiments addressing technical details and metric robustness, critical concerns regarding the attribution of gains (DINO ablation) and comprehensive evaluation against recent baselines remain inadequately addressed.

**Reviewer Scores:**

Reviewer Vx6s: 6 $\rightarrow$ 6:

Efficiency: Authors proved the 40-epoch training is "lightweight" (<6 hours on 8 GPUs), nullifying the reviewer's main concern about compute cost.

Generalization: Table B confirmed the tokenizer works across different generators (generator-agnostic), turning a "Question" into a "Strength."

Data Completeness: Authors provided the requested PSNR/SSIM metrics and DINO loss ablations, satisfying the demand for empirical depth.

Clarity: The explanation of pFID vs. rFID resolved the reviewer’s confusion regarding evaluation metrics.

Reviewer 3zmd: 4 $\rightarrow$ 4:

DINO Necessity: The reviewer felt DINO was treated as a "default choice" without explicit justification. Although the authors provided ablations, the reviewer's initial stance was that the contribution was only "fair" (2/5), suggesting they may still view the method as an incremental combination of existing techniques.

Trade-off Skepticism: The reviewer highlighted a "critical trade-off" between reconstruction and generation. Even with the authors' "annealing" explanation, a reviewer this certain may remain unconvinced that the degradation is fully mitigated without seeing the revised paper's full "thorough analysis".


Reviewer ugEE : 2 $\rightarrow$ 4:

Concerns Addressed: The reviewer explicitly stated that clarifications on loss components and t-SNE controls addressed their earlier concerns.

Independent Gains: Table A proved latent perturbation works without DINO, satisfying the requirement for better attribution.

Efficiency: Authors clarified post-training is very cheap (<6 hours), making it a more "practical solution" than the reviewer initially thought.

Remaining Skepticism: Despite the progress, the reviewer still questioned the practical utility of pFID beyond retrospective analysis, and their initial Contribution rating was only "fair," suggesting they may still view the work as below the acceptance threshold.

Reviewer 3GFk: 4 $\rightarrow$ 4:

Scope Mismatch: The reviewer demanded text-to-image (t2i) experiments on datasets like JourneyDB. The authors argued t2i is "not the primary scope," which, while technically valid, leaves the reviewer's request for broader empirical validation unfulfilled.

Quality Concerns: The reviewer perceived the paper as being "under considerable time pressure" due to typos and formatting issues. Although authors promised fixes, first impressions regarding "polish" are difficult to reverse without a full manuscript revision.

---

### Decision · Program_Chairs · 2026-01-26

Reject